# Structural and functional insights into the delivery of a bacterial Rhs pore-forming toxin to the membrane

Amaia González-Magaña [1,2,6], Igor Tascón [1,3,6], Jon Altuna-Alvarez [1], María Queralt-Martín [4], Jake Colautti[5], Carmen Velázquez [1,2], Maialen Zabala[1,2], Jessica Rojas-Palomino [4], Marité Cárdenas [1,3], Antonio Alcaraz [4], John C. Whitney [5], Iban Ubarretxena-Belandia [1,3] ✉ & David Albesa-Jové [1,2,3] ✉

Bacterial competition is a significant driver of toxin polymorphism, which allows continual compensatory evolution between toxins and the resistance developed to overcome their activity. Bacterial Rearrangement hot spot (Rhs) proteins represent a widespread example of toxin polymorphism. Here, we present the 2.45 Å cryo-electron microscopy structure of Tse5, an Rhs protein central to *Pseudomonas aeruginosa* type VI secretion system-mediated bacterial competition. This structural insight, coupled with an extensive array of biophysical and genetic investigations, unravels the multifaceted functional mechanisms of Tse5. The data suggest that interfacial Tse5-membrane binding delivers its encapsulated pore-forming toxin fragment to the target bacterial membrane, where it assembles pores that cause cell depolarisation and, ultimately, bacterial death.

Biological competition drives the evolution of diverse families of polymorphic toxins[1–12], including specialised type VI secretion system (T6SS) toxins and Rearrangement hot spot (Rhs) toxins. Rhs polymorphic toxins contain tyrosine/aspartate repeats (YD-repeats)[13] that assemble into a β-sheet that spirals to form a barrel-like/cocoon structure[1,2,14–16]. Proteins containing YD-repeats have acquired a high degree of functional diversity throughout evolution and are distributed within bacteria, archaea, and eukaryotes[1–3,17–23]. Examples of bacterial YD-repeat proteins that mediate intercellular competition include Gram-negative T6SS-associated Rhs toxins, insecticidal toxin complexes (Tc), and the distantly related Gram-positive wall-associated protein A (WapA)[1,2,9,22–24].

Bacterial Rhs toxins have a three-domain architecture: an N-terminal domain of variable composition is responsible for directing the toxins towards a range of secretory pathways, including the type 2 secretion system (T2SS), type 6 secretion system (T6SS), or type 7 secretion system (T7SS)[4]. A central domain containing numerous YD-repeats ends in a conserved aspartyl protease active site defined by the DPXGX$_{18}$DPXG motif, while a highly variable C-terminal toxin domain (Rhs-CT) contributes to the functional diversity of these toxins[13,19]. This polymorphism is thought to be evolutionarily decoupled and acquired by homologous recombination[20]. Structural insight into bacterial Rhs toxins is based on two T6SS-associated toxins, Rhs1 and RhsA[1,2]. In both cases, the Rhs-CT fragments are auto-proteolyzed by the conserved Rhs aspartyl protease but remain encapsulated inside the Rhs barrel, presumably until they are delivered into their target cells. The C-terminal toxin domain of Rhs1, named Tre23, is toxic to *Escherichia coli* by inhibiting protein translation through ADP-ribosylation of the 23 S ribosomal RNA[25]. Based on sequence homology to endonucleases, it is predicted that RhsA targets DNA molecules[26]. While the structures

[1]Instituto Biofisika (CSIC, UPV/EHU), Fundación Biofísica Bizkaia/Biofisika Bizkaia Fundazioa (FBB), 48940 Leioa, Spain. [2]Departamento de Bioquímica y Biología Molecular, University of the Basque Country, 48940 Leioa, Spain. [3]Ikerbasque, Basque Foundation for Science, 48013 Bilbao, Spain. [4]Laboratory of Molecular Biophysics, Department of Physics, University Jaume I, 12071 Castellón, Spain. [5]Department of Biochemistry and Biomedical Sciences, Michael DeGroote Institute for Infectious Disease Research, and David Braley Centre for Antibiotic Discovery, McMaster University, Hamilton, Canada. [6]These authors contributed equally: Amaia González-Magaña, Igor Tascón. ✉e-mail: ivan.ubarrechena@ehu.eus; david.albesa@ehu.eus

of RhsA and Rhs1 provide valuable insight into their molecular function, the mechanisms by which their encapsulated toxin domains are delivered to the target cell cytoplasm remain largely unknown.

The T6SS is a contractile secretion system that assembles inside many Gram-negative bacteria and injects effector proteins upon contacting neighbouring cells, providing a fitness advantage that allows the bacteria to compete for space and resources[27,28]. Recently, we identified the molecular function of Tse5-CT, which is the toxic C-terminal fragment of the T6SS exported effector Tse5 (PA2684)[29]. Tse5 is produced by *Pseudomonas aeruginosa* and associates with the VgrG1c (also known as VgrG4) spike complex for secretion by the T6SS[7,30].

Tse5-CT is toxic when expressed in the cytoplasm of *E. coli*[7] and when directed to its periplasm[30]. Furthermore, we showed that ectopic expression in *Pseudomonas putida* EM383 cells of Tse5-CT or a variant encoding for the PelB leader sequence changes their membrane potential, causing membrane depolarisation and bacterial death[29]. Tse5-CT can spontaneously partition into the hydrophobic core of a lipid monolayer when introduced into a polar buffer. Furthermore, when reconstituted on planar lipid bilayers, Tse5-CT forms ion-selective membrane pores characterised by relatively stable currents, which we attribute to the action of proteolipidic structures[29].

In the present study, we have determined the 2.45 Å Cryo-EM structure of Tse5, revealing that it is an Rhs toxin processed into three polypeptide fragments that remain associated with one another via protein-protein interactions. The central Tse5 fragment assembles a shell-like/cocoon structure (Tse5-Shell) that functions as a chaperone to encapsulate the Tse5-CT fragment, allowing the transport of this integral-membrane protein toxin from *P. aeruginosa* to the periplasm of target bacteria. Remarkably, our biophysical data using artificial bilayers demonstrate that Tse5 can independently bind to the surface of model membranes, where it delivers the Tse5-CT fragment. Furthermore, we propose a model of interfacial membrane binding to rationalise the initial step in Tse5-membrane recognition, where an amphipathic surface in Tse5-Shell develops surface interactions with the membrane.

## Results and discussion

### Tse5 structural insight from its 2.45 Å cryo-EM structure

To provide insight into the molecular mechanism of Tse5, we determined its cryo-electron microscopy (cryo-EM) structure to 2.45 Å resolution (Fig. 1) (see Methods, Supplementary Information, Supplementary Fig. 1, 2, and Supplementary Table 1 for details). The cryo-EM density map was of high quality, allowing the ab initio model building of the structure except for residues 1–29, 873–909 and 1196–1317, as a result of their structural flexibility or disorder (Fig. 1b, Supplementary Fig. 1).

The cryo-EM structure revealed Tse5 is fragmented into three peptides (Fig. 1a). Mass spectrometry confirmed this observation and N-terminal sequencing identified N- and C-terminal proteolytic cleavage sites in between amino acid residues Lys47-Pro48 and Leu1168-Ile1169, respectively (see Methods and Supplementary Notes 2–5 for details). The 126-kDa central fragment forms the hollow shell-like/cocoon structure (Tse5-Shell; residues 48–1168), the 5-kDa N-terminal fragment (Tse5-NT; residues 1–47) anchors to the Tse5-Shell, and the 16-kDa C-terminal fragment (Tse5-CT) corresponds to the pore-forming toxin (residues 1169–1317)[29].

Tse5-Shell can be divided into three domains. One domain consists of a barrel-like structure that is assembled from an anti-parallel β-sheet that spirals anticlockwise along the central axis (Fig. 1c, d, and Supplementary Fig. 3b). The remaining two domains correspond to two plugs that close the barrel's apertures (Fig. 1e, f; N-terminal and C-terminal plugs contain residues 48–124 and 1092–1168, respectively). The barrel-like structure of Tse5 assembles from 40 β-hairpins, having the consensus sequence YDXXGRLV, similar to the previously

defined YD-repeats for *E. coli* Rhs proteins (YDXXGRL[I/T])[13], but with significant sequence variability. The DXXGR motif creates the turn of each β-hairpin, with the aspartic acid side chain and backbone hydrogen bonding to the glycine and arginine backbones (blue in Supplementary Fig. 3b). The glycine residue is highly conserved, allowing the torsional freedom necessary for the turn. The aspartic acid and arginine residues are frequently replaced by asparagine, serine, leucine, and glutamine residues, without significantly affecting the hydrogen bonding interactions between β-strands. The tyrosine residue in the YD-repeat faces the inner cavity, and hydrogen bonds to the leucine residue of the preceding YD-repeat. This hydrogen bonding pattern is highly conserved even though other bulky residues like histidine, arginine, tryptophan, or phenylalanine commonly substitute the tyrosine residue (green in Supplementary Fig. 3b). The leucine residue (shown in orange) at the C-terminal end of the YD-repeat lies inside the barrel and is most frequently replaced by an arginine residue. The last residue (shown in yellow) in the Tse5 YD-repeat is solvent exposed, and it is commonly found as valine, leucine, isoleucine, alanine or threonine residues.

The C-terminal end of Tse5-NT (residues 30–47) is anchored to the inner cavity of Tse5-Shell by protein-protein interactions and further stabilised by several intra-molecular hydrogen bonds. Most of these interactions are mediated by the Tse5-Shell's N-terminal plug (residues 48–124). Gln114, Arg116 and Gly119 of the Tse5-Shell's N-terminal plug are essential to establish a network of electrostatic interactions, including hydrogen bonds and salt bridges, with residues of the Tse5-NT fragment. Residue Gln114 interacts with Ser39, Cys41 and Arg37, while Gly119 contacts Leu34, and Arg116 interacts with Asp36 and Ser39, Val32, Gly33 and Leu34. Moreover, residues Arg117, Ile118, Phe120 and Pro121 of the N-terminal plug and Ser131, Ser133, Glu134 and Gln363, localised at the N-terminal aperture of Tse5-shell, thread the N-terminal to the inner cavity mainly through hydrophobic interactions. The whole fragment is further stabilised with several Van der Waals and hydrophobic interactions either with the N-terminal plug or residues of the inner wall of the cavity (Supplementary Fig. 4). Interestingly, Gly361, Gly362, Arg334, Gln326, Glu328 and Trp355 are heavily involved in anchoring Tse5-NT residues next to the cleavage site. Supplementary Table 2 and Supplementary Fig. 4 describe all the interactions between Tse5-Shell and Tse5-NT fragment calculated using the Mapiya server[31].

Tse5-Shell's cavity volume is ~32,000 Å³, which is similar to RhsA and Rhs1 cavity volumes, and sufficient to accommodate the 16-kDa Tse5-CT (RhsA and Rhs1 cavity volumes are ~32,200 Å³ and ~35,000 Å³, respectively, and both have a ~15-kDa C-terminal toxin fragment). Consistent with Tse5-CT being encapsulated within Tse5, Small Angle X-ray Scattering (SAXS) experiments determined comparable $R_g$ values for a Tse5-CT deletion variant (Tse5-ΔCT) and Tse5 (Supplementary Table 3). For Tse5-CT (residues 1169–1317), we could only assign the first 27 N-terminal residues (residues 1169–1195) inside the Tse5-Shell cavity. The cryo-EM density inside the Tse5-Shell cavity for the remaining C-terminal 1196–1317 residues of Tse5-CT was too weak (Supplementary Fig. 3a). Such weak density is consistent with the encapsulated Tse5-CT remaining mostly flexible or disordered, in agreement with cryo-EM maps of Rhs1[1] and RhsA[2] where only 10% and 23% of their encapsulated C-terminal toxin fragments could be resolved, respectively.

### Tse5 delivers its encapsulated Tse5-CT toxin to target membranes

To evaluate the capacity of Tse5 to insert in biological membranes and form pores, we measured its ability to spontaneously (i) partition in lipid monolayers and (ii) induce currents in lipid bilayers (see Methods for details). We evaluated membrane insertion using the Langmuir–Blodgett balance[32]. This technique records the insertion of a protein into a monolayer as an increase in lateral pressure (Δ$\Pi$) from an

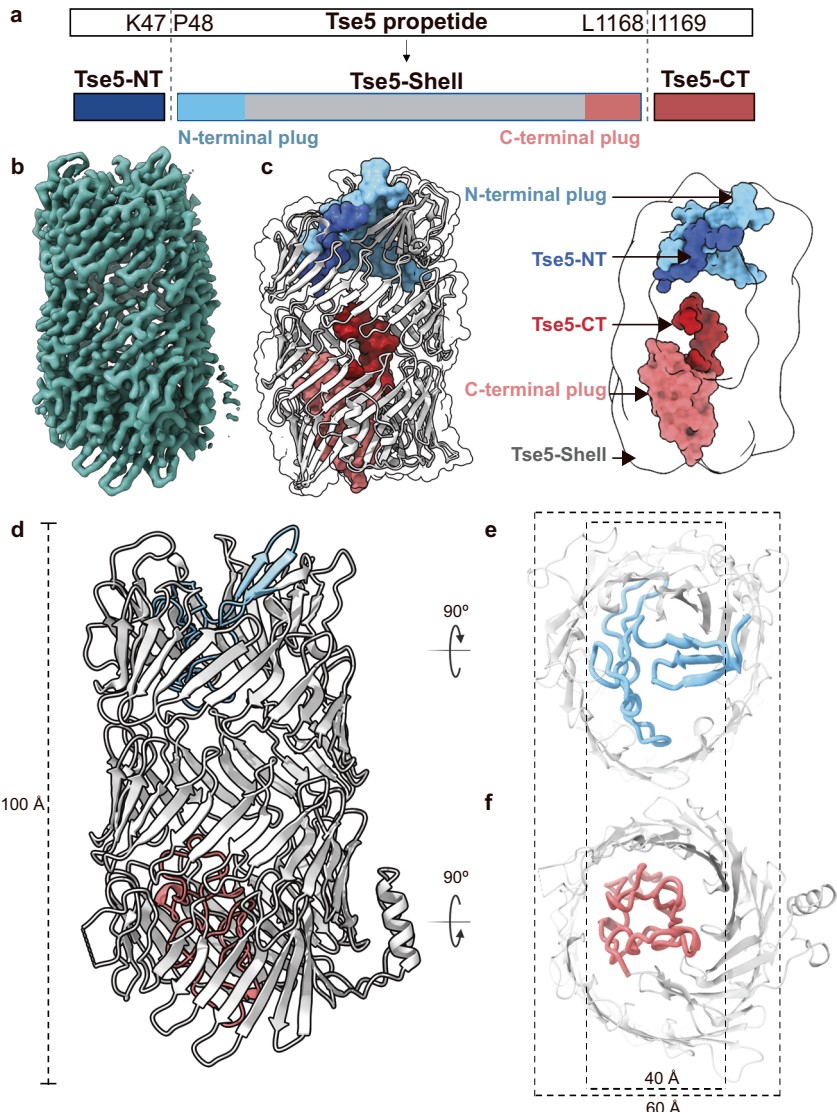

**Fig. 1 | 2.45 Å cryo-EM structure of *Pseudomonas aeruginosa* Tse5. a** Schematic representation of Tse5 propetide and its mature form, indicating it is fragmented in three peptides: a 126-kDa central fragment (Tse5-Shell; residues P48-L1168), a 5-kDa N-terminal fragment (Tse5-NT; residues M1-K47), and a 16-kDa pore-forming toxin (residues L1169-Q1317), previously named Tse5-CT. Representation not to scale. **b** Sharpened Tse5 Cryo-EM map obtained with cryoSPARC at 2.45 Å (contour level 0.196) **c** Structural organisation of Tse5. On the left, the Tse5-Shell is depicted in grey as a cartoon and transparent surface. The residues corresponding to the N-and C-terminal fragments and the N- and C-terminal plug domains are shown in surface representation, coloured in blue, red, soft blue and pink, respectively. On the right, there is a schematic representation of each fragment, showing the Tse5-Shell cavity. To calculate the cavity, Gaussian filtering was applied to the unsharpened map, and the inner volume was subtracted using ChimeraX. **d** Side view of the Tse5-Shell fragment. The barrel-like structure of the Tse5-Shell is shown in grey, while its N-and C-terminal plugs are shown in light blue and pink, respectively. **e** The top view of the Tse5-Shell shows its N-terminal plug. **f** The bottom view of the Tse5-Shell shows its C-terminal plug. The inner and outer diameters of the barrel-like structure are shown.

adjusted initial lateral pressure ($\Pi_0$). Protein insertion decreases as the initial lateral pressure increases until the critical lateral pressure ($\Pi_c$) is reached. At this point, the protein can no longer insert into the monolayer. The lipid packing in the outer monolayer of biological membranes approaches lateral surface pressures between 30 and 35 mN/m[33,34]. Thus, a critical lateral pressure in this range upon protein addition indicates that the protein is spontaneously inserting into the hydrophobic core of the lipid monolayer.

Tse5 insertion into the lipid monolayer yielded a $\Pi c$ near to 35 mN/m (34.74 mN m$^{-1}$, Fig. 2a). Importantly, these results are comparable to those obtained previously for Tse5-CT[29], indicating a similar ability of Tse5 and Tse5-CT to spontaneously partition into the hydrophobic core of a lipid monolayer. Therefore, much like Tse5-CT[29], Tse5 might have the ability to form pores that transport ions across the membrane. To explore this idea in greater detail, we employed a modified solvent-free Montal-Mueller technique[35]. This method involves the creation of an artificial lipid bilayer structure spanning a small orifice embedded within an insulating Teflon barrier that divides two buffer-containing chambers. In doing so, the technique allows monitoring of ion channel activity upon protein insertion into the lipid bilayer. The results of this experiment indicate that Tse5 can form selective ion channels in lipid bilayers with a mild preference for cations, and an ion permeability ratio ($P_K + / P_{Cl}^-$) of $3.8 \pm 0.8$ ($n = 12$) under a concentration gradient of 250/50 mM KCl (Fig. 2b). Remarkably, these results are comparable to those obtained previously for Tse5-CT[29].

Furthermore, we tested Tse5 and Tse5-CT ability to form pores when reversing the salt concentration gradient. To do so, we

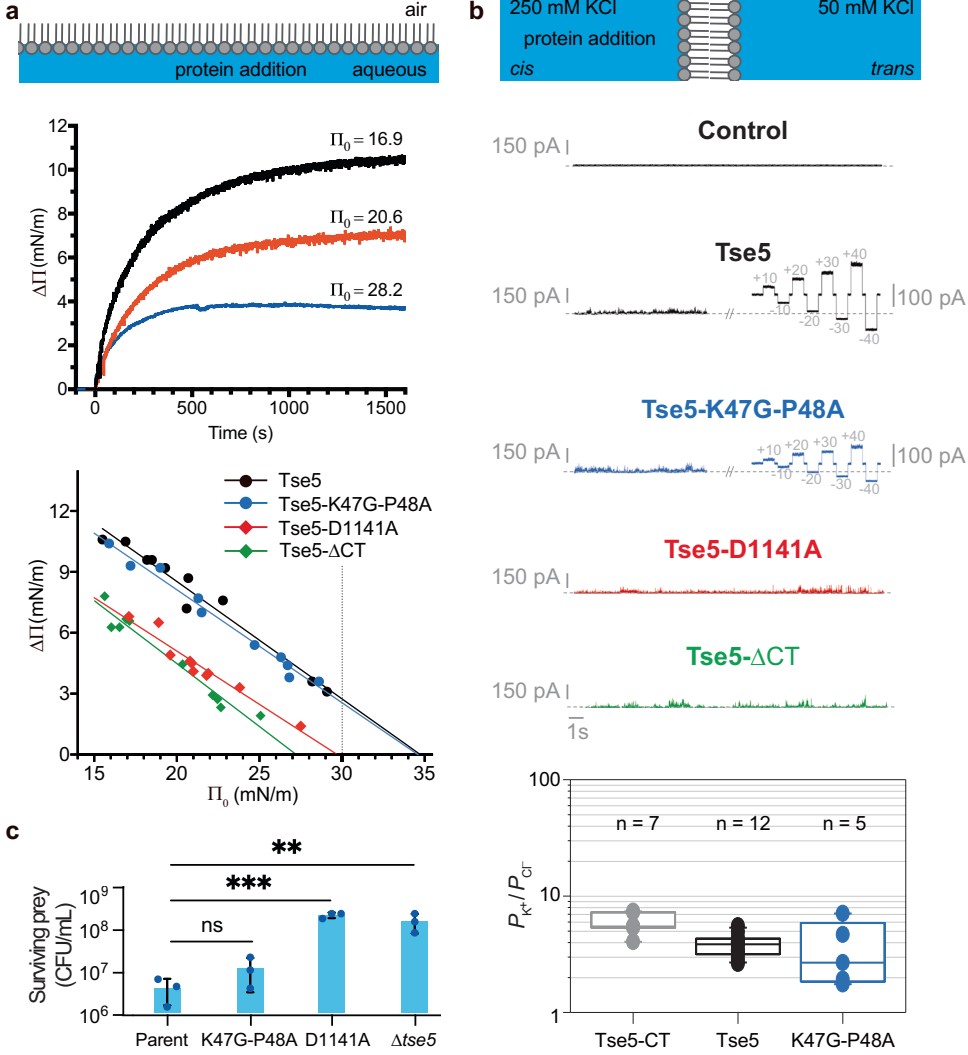

**Fig. 2 | Tse5 membrane insertion and pore-formation requires cleaved Tse5-CT.**
**a** The top panel shows a phospholipid monolayer assembly in a typical Langmuir–Blodgett balance experiment. A monolayer is formed at the air-buffer interface, and the protein is introduced into the buffer. Changes in the monolayer's lateral pressure are recorded over time until reaching the equilibrium (middle panel). The ability of Tse5 (black), the Tse5-ΔCT (green), Tse5-K47G-P48A (blue) and Tse5-D1141A (red) variants to insert into the hydrophobic core of lipid mono-layers spontaneously is calculated by extrapolating the fitted curve to $\Delta\Pi = 0$. The critical lateral pressure ($\Pi_c$) values for Tse5, Tse5-K47G-P48A, Tse5-D1141A, and Tse5-ΔCT are 34.74 mN m$^{-1}$, 34.58 mN m$^{-1}$, 29.05 mN m$^{-1}$, and 27.17 mN m$^{-1}$, respectively. The dotted line indicates the threshold value of lateral pressure consistent with unstressed biological membranes. Each dot corresponds to an independent experiment, representing the lateral pressure increase ($\Delta\Pi$) as a function of initial lateral pressure ($\Pi_0$; $n = 10$ for each protein). **b** The top panel shows a schematic representation of a planar lipid bilayer assembly using the modified solvent-free Montal-Mueller technique. Two chambers (cis/trans) are separated by a lipid bilayer, and a 250/50 mM gradient is generated. Protein is added on the cis side. Representative current traces were obtained before (Control) and after the addition of Tse5 or the variants Tse5-K47G-P48A, Tse5-D1141A and Tse5-ΔCT. All tested variants display current spikes indicative of an interaction between the protein and the lipid bilayer, while stable channels are observed for Tse5 and Tse5-K47G-P48A but not with Tse5-D1141A or Tse5-ΔCT. In all traces, dashed lines indicate zero current levels. The applied voltage was 100 mV for the current spikes, while in the IV curves, it is indicated by grey numbers. The recordings were digitally filtered at 500 Hz using a low-pass 8-pole Bessel filter for better visualisation. The lower panel shows the permeability ratios for the Tse5 C-terminal fragment (Tse5-CT)[29], Tse5 and the Tse5-K47G-P48A variant. The borders of the boxes define the 25th and 75th percentiles, and the line within each box marks the median. Whiskers indicate the standard deviation of the mean. Data are means of 7 (Tse5-CT, grey), 12 (Tse5, black), and 5 (Tse5-K47G-P48A, blue) independent experiments. Solid circles correspond to the individual data points with the minima and maxima being 4.06 and 7.41 for Tse5-CT, 2.63 and 5.56 for Tse5, and 1.75 and 7.10 for Tse5-K47G-P48A **c** In vivo relevance of the Tse5 propeptide cleavage. Bacterial competition assays demonstrate that the cleavage of the Tse5-NT is not essential, while the cleavage of the Tse5-CT is essential for toxicity. Data are represented for the surviving Tse5-susceptible *P. aeruginosa* PAO1 prey strain following 20 h of competition with donor *P. aeruginosa* PAO1 strains with an active T6SS and either expressing the wild-type Tse5 (Parent) or variants Tse5-K47G-P48A (K47G-P48A), Tse5-D1141A (D1141A) or not expressing Tse5 (Δtse5). Each bacterial competition was done in triplicate ($n = 3$). The mean with standard deviation (SD) and the value of each replicate are plotted. Statistical significance was evaluated with the ordinary one-way ANOVA with Dunnett's multiple comparison test. *P*-values of Parent vs K47GP48A, D1141A, and Δtse5 are 0.9898, 0.0006, and 0.0052, respectively. *P*-value > 0.1234 (ns), 0.0332 (*), 0.0021 (**), 0.0002 (***), < 0.0001 (****). Plotting and data analysis were done with GraphPad Prism v9.5.

monitored ion channel activity of Tse5 and Tse5-CT under a concentration gradient of 250/50 mM KCl and 50/250 mM KCl in membranes formed of a DOPE/DOPG mixture (see Methods for details). Interestingly, we easily observed ion channel activity of Tse5 in a 250/50 gradient, and of Tse5-CT in both gradients, but we could not obtain any ion channel activity with Tse5 using a 50/250 gradient, even at high protein concentrations (Supplementary Fig. 5a). Moreover, we investigated the effect of an asymmetry in the lipid composition of the membrane. To that end, we performed experiments with Tse5 or Tse5-CT using bilayers formed of neutral lipid (DOPE) in one monolayer and anionic phospholipid mixture (DOPE/DOPG) in the other monolayer (see Methods for details). Remarkably, we also observed a difference in the behaviour of Tse5 and Tse5-CT. While Tse5-CT could induce stable currents in all cases, we did not observe any ion channel activity with Tse5 when the charged lipid was present on the side of protein insertion and the neutral lipid formed the opposite monolayer. However, in the inverse situation, with the neutral lipid on the side of protein insertion and the charged lipid on the opposite side, Tse5 could still generate some pores (Supplementary Fig. 5b).

These results suggest that Tse5 might react differently depending on the orientation of the electrical potential across the membrane, which is emulated in our bilayer system by assembling an asymmetrically charged membrane or inducing a salt concentration gradient in a symmetric bilayer. The effect in this latter case is attained by screening the lipid negative charges at one side of the membrane more than at the other side. The fact that only Tse5 pore-forming activity is affected by this directionality points to an effect on the delivery or release of Tse5-CT from inside Tse5-Shell. Therefore, our findings suggest that a negative transmembrane potential could inhibit Tse5's ability to deliver its toxic cargo. Given that this potential corresponds to the one felt from the cytoplasmic side of the cell membrane, we postulate that Tse5 preferably acts from the periplasmic side. This differs from Tse5-CT activity, which is toxic when expressed in the cytoplasm of *E. coli*[7] and when directed to its periplasm[30], causing in both instances membrane depolarisation and bacterial death[29].

## Differential requirement of Tse5-CT and Tse5-NT cleavage for toxin activity

Although Tse5 shares relatively low sequence identity with RhsA and Rhs1 (26.5% and 21.1%, respectively), the folding of their N- and C-terminal plugs are similar, with RMSD values between 1.8–2.9 Å (the RMSD across all protein residues are between 5.1–6.0 Å; Fig. 3a). Furthermore, the Tse5-Shell C-terminal plug contains a DPXGX$_{19}$DPXG motif (Fig. 3b) similar to the aspartyl protease motif (DPXGX$_{18}$DPXG) found in many Rhs proteins from the Enterobacteriaceae family[20].

The available data on Rhs proteins suggests a conserved auto-proteolytic mechanism that cleaves the C-terminal toxin fragments. A conserved aspartyl protease active site mediates this cleavage, and single-point variants of these conserved aspartic residues in T6SS Rhs homologues abolishes autoproteolytic activity[1,24,36]. We found that this is also true for Tse5 as point mutants Tse5-D1141A and Tse5-D1164A cannot cleave at the Leu1168-Ile1169 cleavage site (Fig. 3c, d), supporting the idea that this plug region of Tse5 is an aspartyl protease domain.

Studies of *Acidovorax citrulli* RhsB[36] and *Aeromonas dhakensis* TseI[24], which are other characterised T6SS Rhs effectors, have examined their auto-proteolytic activity using an in vitro protein synthesis, which consists of purified cell-free transcription-translation protein components. The results from this experiment indicate that RhsB and TseI spontaneously self-cleave and therefore suggest that auto-proteolysis occurs before secretion and independently of T6SS functionality.

To test if the cleavage between residues Leu1168−Ile1169 is required to produce an active pore-forming toxin, we compared the capacity of wild-type Tse5 and the Tse5-D1141A variant to (i) insert into

model membranes spontaneously and (ii) form membrane pores (see Methods for details).

SAXS analysis indicates Tse5-D1141A possesses similar overall structural parameters to wild-type Tse5, having comparable values of their radius of gyration ($R_g$), maximum distance ($d_{max}$) or estimated molecular weight (MW) (Supplementary Table 3). Nevertheless, its capacity to insert into the hydrophobic core of lipid monolayers is severely reduced compared to the wild-type protein, as indicated by its reduced critical lateral pressure value ($\Pi_c$ of Tse5-D1141A and Tse5 are 29.05 and 34.74 mN m$^{-1}$, respectively; Fig. 2a).

In addition, Tse5-D1141A shows fewer permeabilisation events than Tse5, with almost all observed events being unstable asymmetric currents−which we ascribe to the process of frustrated pore assembly attempts−or small current spikes and only residually in the form of stable ohmic channels (Fig. 2b). Therefore, we conclude that the D1141A substitution abrogates the formation of stable membrane pores.

We next conducted bacterial competition assays to evaluate the importance of Tse5-CT auto-proteolysis for the function of this toxin in vivo (Fig. 2c). To this end, we evaluated the competitiveness of *P. aeruginosa* donor strains expressing either wild-type Tse5 or Tse5-D1141A against a recipient lacking both *tse5* and the gene encoding its cognate immunity protein, *tsi5* (Δ*tse5* Δ*tsi5*). As expected based on our in vitro results, the donor strain expressing Tse5-D1141A could not outcompete Tse5-sensitive recipients. By contrast, the strain expressing the wild-type toxin did outcompete the Tse5-sensitive recipients (Fig. 2c). Furthermore, we found that the reduction in co-culture fitness of the aspartyl protease variant is comparable to that of a donor strain lacking the *tse5* gene altogether (Δ*tse5*), indicating that the D1141 is essential for Tse5 function during bacterial competition.

The N-terminal plug that caps the barrel's N-terminal aperture corresponds to a Domain of Unknown Function (DUF6531; Figs. 1c, e, 3a). Interestingly, this N-terminal plug anchors the Tse5-NT fragment inside Tse5-Shell via protein-protein interactions (Supplementary Fig. 4). Compared to the C-terminal autoproteolysis event, which is well-documented among Rhs proteins, less is known about the mechanism and significance of the N-terminal cleavage. A D288N point mutant in the Rhs1 toxin inhibits the cleavage of its N-terminal fragment, which led the authors to suggest that the Asp288 could be a catalytic residue involved in auto-proteolysis[1]. The corresponding position in Tse5 and RhsA is not conserved; instead, residues Lys47 or His304 are present at this position, respectively (Fig. 3b). Mutation of two glutamic acids in TseI (E428, E429) also abrogates N-terminal cleavage. However, these residues are also not conserved in Tse5 and RhsA (Fig. 3b). Therefore, the mechanism by which the N-terminus is cleaved in Tse5 would seem to diverge from the mechanism employed by Rhs1 and TseI. Nonetheless, we were able to abrogate N-terminal processing by mutating the cleavage site residues Lys47 and Pro48 to glycine and alanine residues, respectively (K47G and P48A; Fig. 3c, d).

To ensure that Tse5-K47G-P48A possesses similar overall structural parameters to wild-type Tse5, we performed SAXS analysis and found no major differences (Supplementary Table 3). Next, to test if the cleavage of the Tse5-NT is required to yield an active toxin, we measured the capacity of the Tse5-K47G-P48A variant to (i) insert into model membranes spontaneously and (ii) form membrane pores (see Methods for details). The results of these experiments indicate that this variant can insert into the hydrophobic core of a lipid monolayer (Fig. 2a), having $\Pi_c$ values comparable to those obtained for Tse5 ($\Pi_c$ of Tse5-K47G-P48A and Tse5 are 34.58 and 34.74 mN m$^{-1}$, respectively). Furthermore, the Tse5-K47G-P48A variant showed intense ion channel activity with conductive levels comparable to that found for Tse5 (Fig. 2b). These pores had a multi-ionic character with a mild preference for cations with an ion permeability ratio ($P_{K^+}/P_{Cl^-}$) of $3.63 \pm 2.26$ ($n = 5$), comparable to the selectivity obtained for Tse5, and previously reported for Tse5-CT tested under

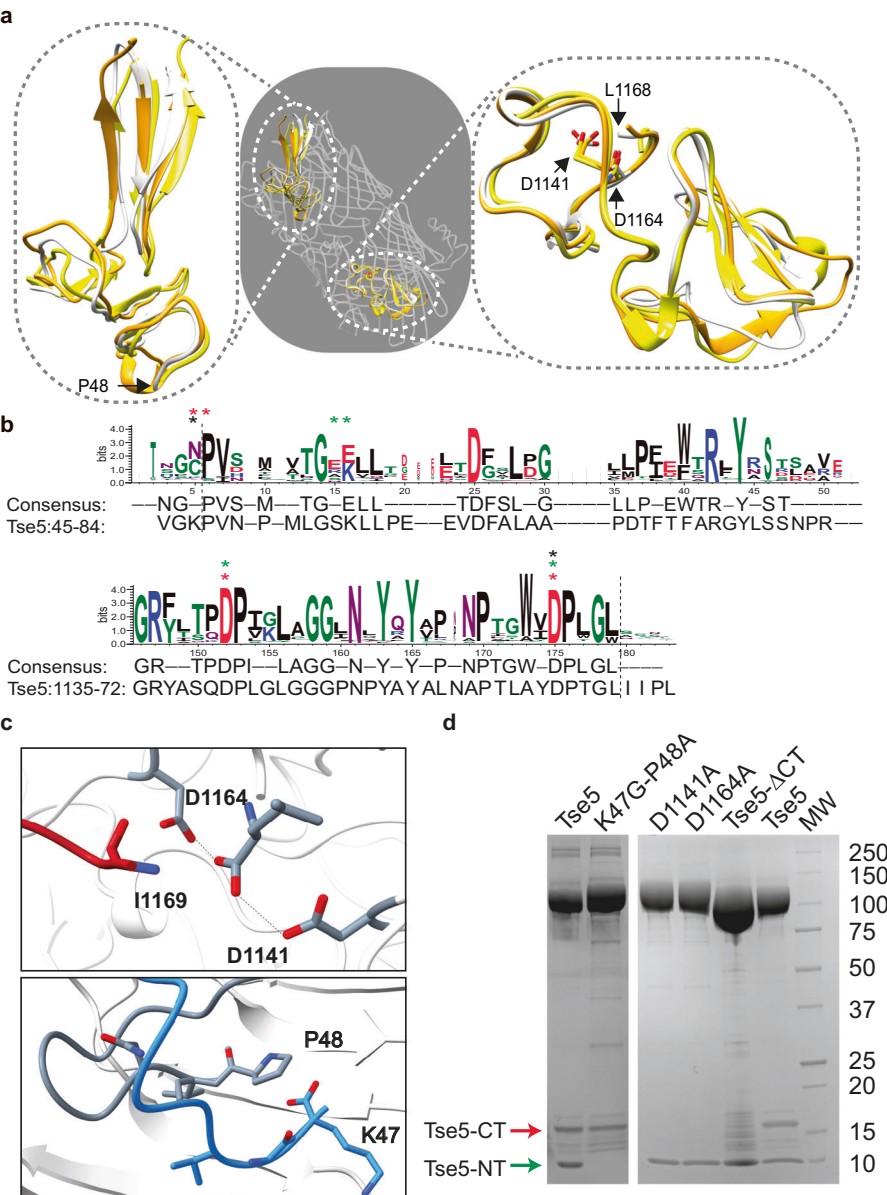

**Fig. 3 | Differential requirement of Tse5-CT and Tse5-NT cleavage for toxin activity. a** Structural superposition of Tse5 (grey), RhsA (orange), and Rhs1 (yellow) N- and C-terminal plugs. N- and C- terminal residues (P48 and L1168) in Tse5's N- and C-terminal plugs are indicated. Conserved aspartic residues in the aspartyl protease motif within the C-terminal plug are also indicated (D1141 and D1164). **b** Conservation in the N- and C-terminus of Tse5-Shell. Sequence logo around Tse5 residues 45–84 (top) and 1135–1172 (below) generated from an alignment of 2225 sequences from *Pseudomonas* containing the N-terminal plug domain of unknown function (DUF6531; InterPro entry PF20148). Tse5 cleavage sites are indicated with a dotted line. Mutated positions that inhibit the N-terminal and C-terminal cleavage in Rhs1 (black *), RhsA (red *), TseI (green *) or Tse5 (red *) are starred. Sequence conservation consensus and the Tse5 sequence are provided below. Sequence logo generated by WebLogo[78]. **c** N- and C-terminal cleavage sites. The Tse5-Shell is depicted in grey/white cartoon. In the upper panel, cleavage site of the C-terminus (between L1168 and I1169) and the catalytic aspartic residues (D1141 and D1164) are shown. The lower panel illustrates the N-terminal cleavage site (between K47 and P48). **d** Representative SDS-PAGE of purified Tse5 and variants. Similar results were obtained for >10 protein batches purified to homogeneity. The Tse5-NT (green arrow) and Tse5-CT (red arrow) fragments are visible in the pure Tse5. Mutation of the N-terminal cleavage site (Tse5-K47G-P48A) abrogates the scission of the 10-kDa Tse5-NT. Mutation of the catalytic aspartic residues (D1141A and D1164A) abrogates the scission of the -16-kDa Tse5-CT.

the same conditions[29]. Consistent with these biophysical findings, a *P. aeruginosa* PAO1 donor strain expressing Tse5-K47G-P48A exhibits a competitive advantage against a Tse5-susceptible recipient strain that is comparable to that of a donor strain expressing wild-type Tse5 (Fig. 2c).

These are significant findings because, although experimental confirmation lacks for RhsA, our results differ from Rhs1, and TseI, where the N-terminal cleavage is essential for their activity[1,24], suggesting an evolutionary divergence in bacterial Rhs toxins that might

be associated with a loss/gain of function of the N-terminal fragments. While the Tse5-NT functionality would seem to be reduced to VgrG spike interaction for the secretion of Tse5[7,30], the N-terminal fragments of Rhs1 (Rhs1-NT), TseI (TseI-NT), and RhsA (RhsA-NT) likely contain additional functions. The 29-kDa Rhs1-NT, 45-kDa TseI, and 25-kDa RhsA-NT include predicted transmembrane regions, which in some cases are hypothesised to insert into the inner membrane of the target cell to allow the trafficking of their C-terminal toxic fragments into the cytosol[1,37]. Perhaps this possible

membrane targeting specialisation of Rhs1-NT and TseI-NT explains why their cleavage is essential.

## Interfacial Tse5-membrane binding delivers Tse5-CT toxin to the target membrane

A major difference between Tse5 and Rhs1 or RhsA resides in the functional variability between their C-terminal toxin fragments. The C-terminal toxin fragments of Rhs1 and RhsA exert their toxicity in the cytoplasm of their bacterial target. In particular, the 15-kDa Rhs1-CT, named Tre23, is toxic to *E. coli* by inhibiting protein translation through ADP-ribosylation of the 23 S ribosomal RNA[25] while the 15-kDa RhsA-CT is predicted to target DNA molecules[26].

Importantly, Rhs1-CT and RhsA-CT are soluble enzymes, and their corresponding Rhs effectors presumable evolved to translocate them into the cytoplasm of target bacteria upon delivery into the periplasm by the T6SS. It was previously suggested that their N-terminal fragments mediate this translocation by the action of several predicted transmembrane regions coded within their N-terminal fragments[1,2]. Thus, it would seem that Rhs1-NT and RhsA-NT have acquired dual functionality, that is, association to their T6SS for secretion and translocation of their C-terminal toxic fragments into the cytosol of target bacteria. In contrast, Tse5-CT inserts into the cytoplasmic membrane to exert its toxic effect[29]. Furthermore, its 5-kDa Tse5-NT only contains a PAAR-like domain necessary for secretion. Therefore, we raised the question of whether, in the absence of a membrane-binding module in Tse5-NT, Tse5-Shell is responsible for binding to target membranes to deliver its toxic cargo.

To address this question, we compared the membrane-interaction properties of a Tse5-CT deletion variant (Tse5-ΔCT) with Tse5 and Tse5-CT. Tse5-ΔCT shows a dramatic reduction of the critical lateral pressure ($\Pi_c = 27$ mN m$^{-1}$; Fig. 2b). This decrease indicates Tse5-ΔCT is not able to insert into the hydrophobic core of the lipid monolayer, although it does not discard interfacial membrane binding, given that values in this range have been reported for well-characterized peripheral membrane proteins such as mitochondrial creatine kinase, glycolipid transfer protein, and A2 phospholipases[34].

In line with the above result, Tse5-ΔCT cannot form ion channels in lipid bilayers, demonstrating that Tse5-CT is essential for the pore-forming activity of Tse5. However, occasional current spikes reveal that Tse5-ΔCT interacts dynamically with the planar lipid bilayer (Fig. 2b).

To further understand the interaction of Tse5-ΔCT with the membrane, we compared its membrane-binding properties with Tse5 and Tse5-D1141A using Quartz Crystal Microbalance with Dissipation (QCM-D) monitoring, which is a highly sensitive analytical tool used to monitor changes in the mass and viscoelastic properties of thin films or surfaces in real-time at the nanoscale, widely used for measuring protein adsorption to surfaces[38,39]. Adding Tse5, Tse5-D1141A, or Tse5-ΔCT to preformed supported lipid bilayers (Fig. 4a) induced an increase in the mass of the resonator, detectable by a decrease in frequency ($\Delta f$) and an increase in dissipation ($\Delta D$) (Fig. 4b). This indicates that Tse5 and variants can reach the bilayer surface and interact with the supported lipid bilayer by adsorbing to it, with the Tse5-D1141A showing the largest changes in frequency, followed by Tse5-ΔCT, and Tse5.

The time evolution of $\Delta D$ shows additional qualitative trends. Tse5, Tse5-D1141A, and Tse5-ΔCT show slight increases in dissipation over time until reaching a quasi-equilibrium well below 1 ppm. The dissipation (damping) is the sum of all energy losses in the system per oscillation cycle. It can be defined as the energy dissipated per oscillation, divided by the total energy stored in the system[40]. A soft film attached to the quartz crystal is deformed during oscillation, which gives high dissipation. In contrast, a rigid material follows the crystal oscillation without deformation and consequently gives low dissipation, typically well below 1 ppm. Therefore, the similar dissipation

values observed between Tse5, Tse5-D1141A, and Tse5-ΔCT could be qualitatively interpreted as the result of the formation of a rigid film, indicative of a tight protein monolayer formation on top of the lipid bilayer and more protein binding for the variants as compared to Tse5.

Besides the larger extent of adsorption for the variants compared to Tse5, the reversibility of the adsorption differs between Tse5 and the variants: In the case of the variants, the desorption process exhibits a distinct $\Delta D/\Delta f$ slope compared to the adsorption process, whereas for Tse5, the slope for both desorption and adsorption are identical (Fig. 4c). Such behaviours indicate that Tse5 remains largely irreversibly bound to the model membrane, while the variant binding is less stable upon rising with buffer, inducing major changes to the model membrane/protein adsorbed film. These changes could imply, for example, a less tight protein configuration or alterations to the lipid bilayer packing so that the overall adsorbed layer becomes more heterogeneous. QCM-D data alone cannot distinguish between these two processes[39].

These results provide insight into the membrane-binding mechanism of Tse5, indicating that Tse5-ΔCT can bind to the membrane to form a compact layer with the supported lipid bilayer that has similar mechanical properties to the layer assembled with Tse5. Nonetheless, Tse5 binding is largely irreversible, while, upon rinsing with buffer, the binding of Tse5-ΔCT or Tse5-D1141A yields a conformation of the adsorbed protein-lipid bilayer that significantly differs from that of the Tse5-lipid bilayer. This would indicate that the variants can change conformation, probably due to missing interactions with the bilayer core.

## A model of interfacial membrane binding for Tse5-CT toxin delivery

Structural comparison points to three protrusions on the surface of Tse5-Shell not present in Rhs1 and RhsA (regions 141–167, 873–914, and 1031–1062; Fig. 5a). Regions 141–167 and 1031–1062 (*Helical Region 2*) were resolved in the cryo-EM density map and displayed helical content, region 873–914 (*Predicted Helical Region 1*) was not visible in the cryo-EM map, most likely due to structural flexibility or disorder. Nevertheless, it is predicted by AlphaFold[41] to contain a high helical content.

Furthermore, a search with Foldseek[42] identified 13 homologues with known/predicted enzymatic C-terminal toxicities that might be targeting the cytoplasm (see Methods, Supplementary Note 1, Supplementary Data 1 and Supplementary Fig. 6 for details). Remarkably, sequence alignment also detects *Predicted Helical Region 1*, and *Helical Region 2* are absent in these homologues, as well as recognising a *Hydrophobic Patch* (residues 670 and 676–680) between the two helical regions that accumulate mutations (Fig. 5b, Supplementary Fig. 7). Altogether, these surface features contribute to one side of the Tse5-Shell being more amphipathic than the other, which might influence the observed interfacial Tse5-membrane binding (Fig. 5b; Supplementary Fig. 8).

To provide a molecular model for this interfacial membrane binding, we carried out a 1-microsecond Molecular Dynamics (MD) simulation. The initial position of Tse5 in a Gram-negative inner membrane was predicted by the programme PPM 3.0[43], which predicts Tse5-Shell orientates with its *Hydrophobic Patch* at a membrane penetration depth of 0.4 Å, resulting in a calculated membrane binding energy of −3.3 kcal mol$^{-1}$. Similar binding energies have been calculated for F- and I-BAR domains, which are experimentally confirmed peripheral membrane proteins[43].

At the start of the MD simulation, *Predicted Helical Region 1* and *Helical Region 2* are solvent-exposed. As the simulation progresses, they change their conformation to bind to the membrane (Fig. 6a, b). These two helical regions show the highest structural fluctuation of all the protein model. These intrinsic differences are readily observed by their Root Mean Square Fluctuations (RMSF) during the simulation

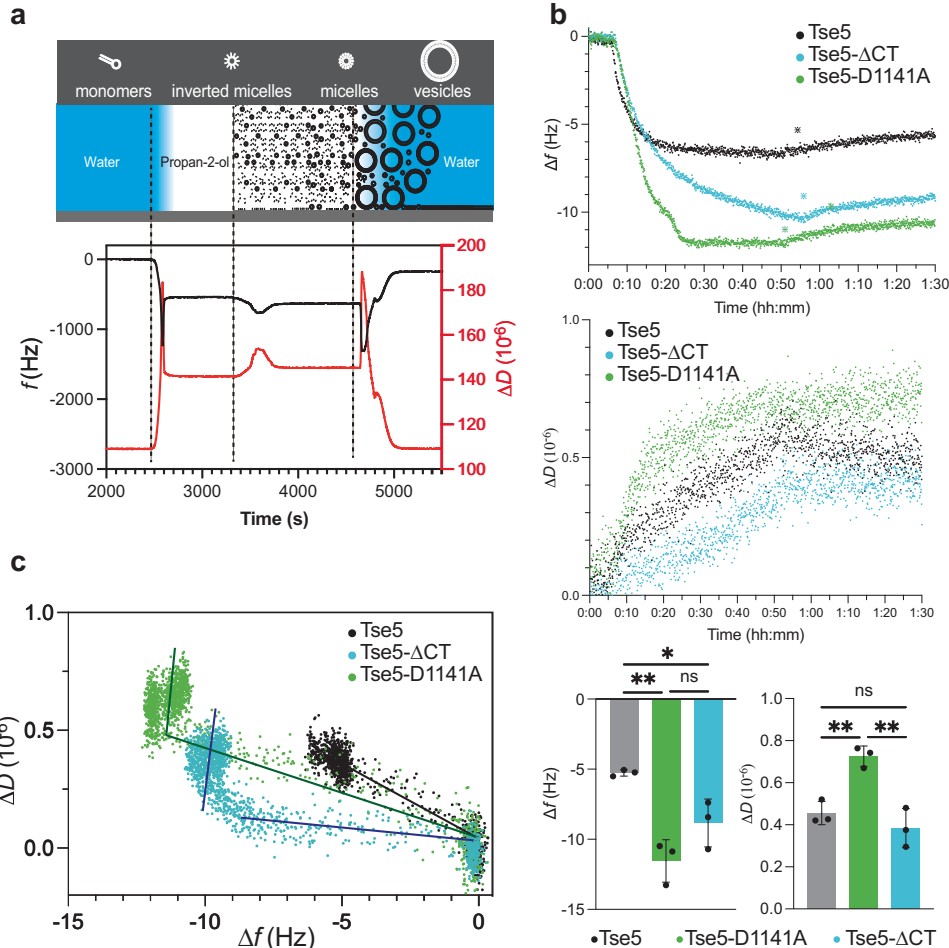

**Fig. 4 | Interfacial Tse5-membrane binding delivers Tse5-CT toxin to the target membrane. a** (Upper panel) Schematic representation of molecular self-assembly of phospholipid molecules using the modified solvent-assisted lipid bilayer (SALB) method. (Lower panel) Selected QCM-D traces showing frequency (black) and energy dissipation (red) changes for 9:1 POPE/POPG bilayer formation. The consecutive steps are divided by dashed vertical lines: water flow, isopropanol exchange, DOPE/DOPG lipid-isopropanol addition and water exchange. **b** Representative QCM-D measurements of changes in frequency (upper graph) and dissipation (central graph) at the seventh overtone upon addition of 1 μM Tse5, Tse5-D1141A or Tse-ΔCT in buffer TBS. QCM-D maximum change in frequency (left bottom graph) and dissipation (right bottom graph) measured for the seventh

overtone after protein addition. Independent experiments were performed in triplicate (*n* = 3). The graphs show the mean values, the standard deviations (SD), and the value of each replicate. Statistical significance was evaluated with the ordinary one-way ANOVA with Tukey's multiple comparison test. *P*-values of Tse5 vs Tse5-D1141A, Tse5 vs Tse5-ΔCT, and Tse5-D1141A vs Tse5-ΔCT are 0.0027, 0.0373, and 0.1011, respectively. *P*-value > 0.1234 (ns), 0.0332 (*), 0.0021 (**), 0.0002 (***), < 0.0001 (****). Plotting and data analysis were done with GraphPad Prism v9.5. **c** Plot showing changes in dissipation as a function of changes in frequency (Δ*D*/Δ*f*) at the seventh overtone upon addition of 1 μM Tse5, Tse5-D1141A or Tse5- ΔCT in buffer TBS. The different Δ*D*/Δ*f* slopes are indicated with lines for visual inspection.

(Fig. 6c). Of note, the high fluctuation of *Predicted Helical Region 1* during the simulation could explain why it was not visible in the cryo-EM map, and provides computational evidence of the relevance of dynamics in interfacial Tse5-membrane binding.

Remarkably, during the entire MD simulation, Tse5 remains bound to the membrane, providing a computational visualisation of the interfacial Tse5-membrane binding. In this model, the protein penetrates the membrane with the *Hydrophobic Patch* at the interface between the lipid's hydrophilic heads and hydrophobic tails in one of the membrane leaflets. This binding also involves the dynamic *Pred. Helical Region 1* and *Helical Region 2* (Supplementary Movie 1).

The 1-μs MD simulation provides a computational model of the interfacial Tse5-membrane binding. This simulation might represent a glimpse into the initial step of toxin delivery into the membrane, which, based on our electrophysiological studies, is the first necessary step for pore formation, a complex process that most likely spans the millisecond-to-second timescale. Therefore, we do not discard the

possibility that other binding modes might be involved in the interfacial Tse5-membrane binding.

## Summary

This study uncovers the 2.45 Å cryo-electron microscopy (cryo-EM) structure of Tse5. Combining the structural insights with detailed biophysical and genetic studies, we provide insight into the functional mechanisms of Tse5, a Rearrangement hotspot (Rhs) protein involved in bacterial competition that is secreted by the type VI secretion system (T6SS) of *Pseudomonas aeruginosa*.

The structure reveals that Tse5 is organised in three polypeptide fragments that remain physically associated through protein-protein interactions. The 5-kDa Tse5-NT fragment contains a PAAR-like domain essential for secretion, with the last 18 residues anchored to the inner cavity the central fragment, which forms a shell-like/cocoon structure termed Tse5-Shell, characterised by diverse tyrosine-aspartate (YD)-repeat sequences. This shell encapsulates the flexible/disordered Tse5-CT fragment, preserving its toxic properties.

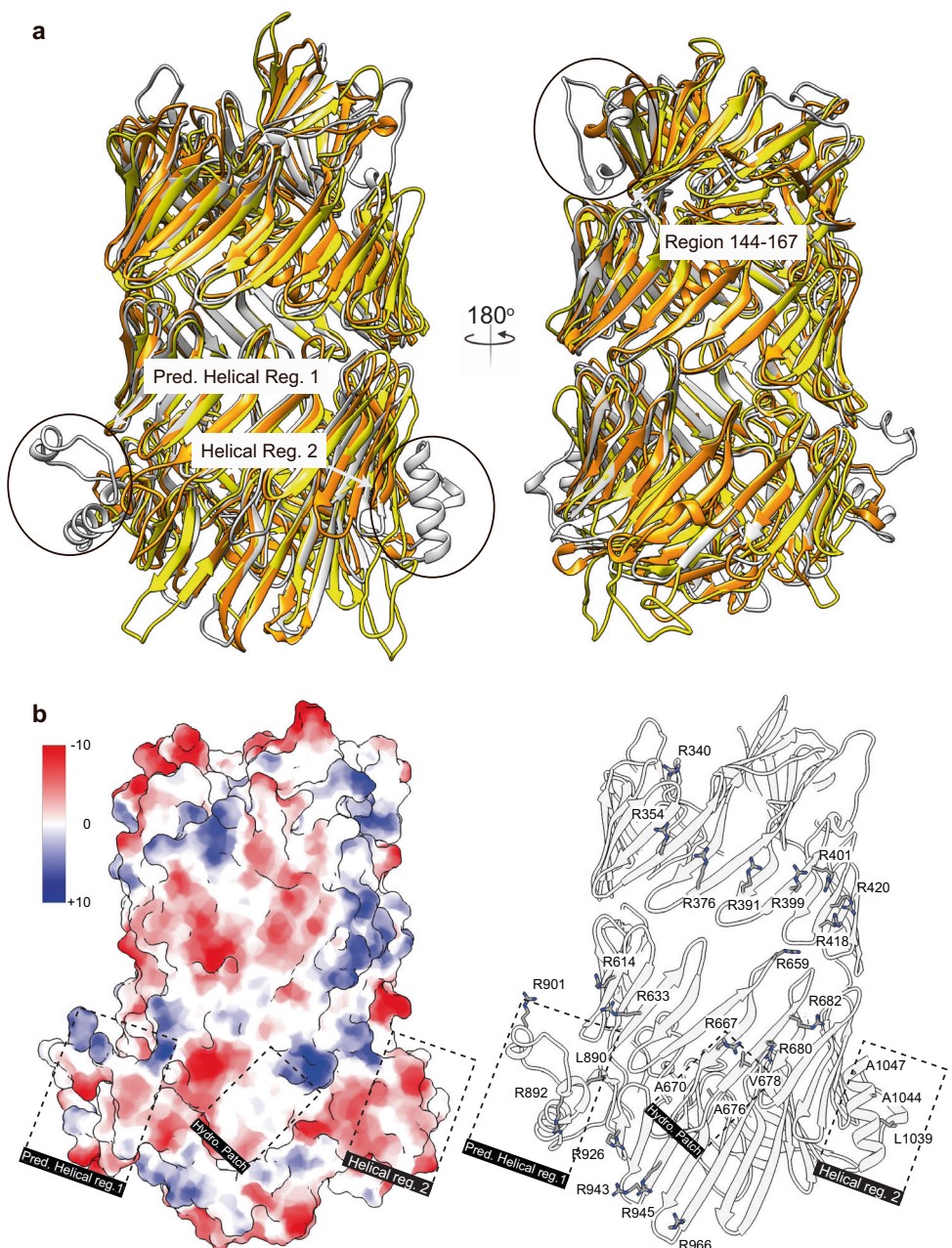

Tse5-Shell's amphipathic surface

**Fig. 5 | Structural comparison of Tse5, RhsA, and Rhs1 reveals unique protrusions on the surface of Tse5-Shell. a** Structural superposition of Tse5 (grey), RhsA (orange), and Rhs1 (yellow). **b** Two views of the Tse5-Shell's amphipathic surface showing the Coulombic electrostatic potential plotted on the solvent-excluded surface (left panel), and residues on the surface contributing to its amphipathic character, including residues in *Pred. Helical Region 1*, *Helical Region 2*, *Hydrophobic Patch*, and other arginine residues on the surface. The colour key on the left indicates the electrostatic surface potential values plotted onto the surface (−10 to +10 kcal/(mol·e⁻) at 298 K). ChimeraX was used to calculate the Coulombic electrostatic potential.

This study demonstrates a differential requirement for cleavage of Tse5-CT and Tse5-NT for toxin activity. Specifically, cleavage of Tse5-CT is essential to activate the toxin, facilitated by the aspartyl protease domain within the Tse5-Shell C-terminal plug. However, the mechanism and functional significance of Tse5-NT cleavage remains enigmatic.

Remarkably, Tse5-NT lacks a membrane-binding module, contrasting with other Rhs proteins. Despite this, a Tse5-CT deletion variant (Tse5-ΔCT) can bind to a supported lipid bilayer, highlighting the importance of interfacial Tse5-membrane interactions in toxin delivery.

Tse5-CT targets the inner membrane of competing gram-negative bacteria, causing cell depolarisation, and spontaneously inserts into artificial membranes, producing ion channel activity with relatively stable currents, which we attribute to the action of proteolipidic pores[29]. There are many pore-forming colicins involved in bacterial competition that share the common functionality of disrupting membrane integrity through pore or channel formation. Nevertheless, they exhibit notable differences in their mechanisms of action. Pore-forming colicins, such as Colicin E1[44] and Colicin A[45], are a group of bacterial toxins produced by various strains of *Escherichia coli*. These toxins belong to the class of bacteriocins and exhibit the characteristic

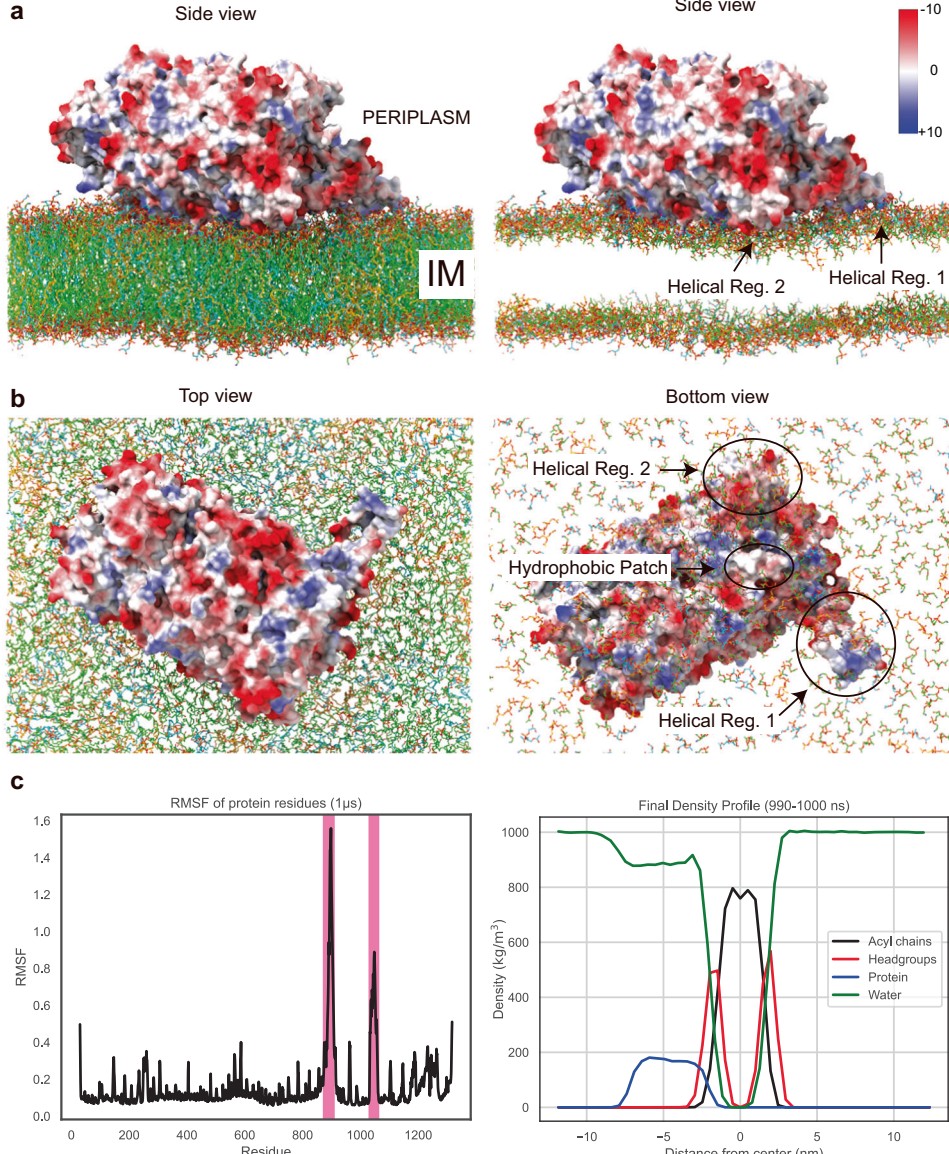

**Fig. 6 | A model of interfacial membrane binding for toxin delivery. a** Side view of the final MD coordinates of the system are shown in the absence of solvent and ions for clarity. In the panel on the right, lipid acyl chains were exchanged to acetyl groups for the same reason. The Coulombic electrostatic potential was plotted onto the solvent-excluded surface of Tse5 with ChimeraX. The colour key on the right indicates the values plotted onto the surface (−10 to +10 kcal/(mol·e⁻) at 298 K). **b** The top and bottom views of the MD coordinates of the system are shown, in which the regions of interest (*Predicted Helical Region 1, Helical Region 2,* and *Hydrophobic Patch*) contact the membrane. **c** Root Mean Square Fluctuations (RMSF) plot for the Cα on each protein residue, computed in the whole μs range of the MD trajectory in GROMACS (left panel). The pink-coloured interval residues belong to the *Predicted Helical Region 1* and *Helical Region 2*. It is visually evident that the residues within those regions fluctuate in position more than the rest of the residues in Tse5. The right panel shows a representational density profile delineating the spatial distributions of molecular components of the simulated system (Tse5 in blue, lipid headgroups in red, acyl chains in black and water in green) for the final 10 ns of the simulation. The densities are measured in kg·m⁻³, taking the central axis of the lipid bilayer as a reference point (zero value).

function of selectively targeting bacterial cells. Colicins display an intricate mechanism of action, which relies on specific receptor recognition and bacterial translocation machinery[46]. Tse5 requires the T6SS for delivery into the periplasm of target bacteria, and once inside this cellular compartment, Tse5 autonomy in membrane insertion sets it apart from colicin.

In conclusion, this work provides valuable insights into the structural organisation and functional mechanisms of Tse5, shedding light on its role in bacterial competition. Furthermore, it highlights the complex interplay of factors governing Tse5-membrane interactions, a critical aspect of its toxin delivery mechanism. Further studies are warranted to elucidate the complete process and potential therapeutic/biotechnological applications.

## Methods

### Tse5, Tse5-ΔCT, Tse5-D1141A, Tse5-D1164A and Tse5-K47G-P48A construct design, protein expression and purification

The *Pseudomonas aeruginosa tse5* gene (PA2684) was synthesised by GenScript and cloned into a pET29a(+) between the NdeI and HindIII restriction sites (pET29a(+)::*9xhis-Tse5*). This construct codes for the protein sequence shown in Supplementary Fig. 10, Supplementary Data 2. The construct contains a 5′ extension encoding for a His-tag and a tobacco etch virus protease cleavage site (ATGGGCAGCAGCC ATCATCATCATCATCATCATCATCATCACAGCAGCGGCGAAAACCTGTA TTTTCAGGGCGGATCC). GenScript derived *tse5* mutants by mutation of the parental vector pET29a(+)::*9xhis-Tse5* (see Supplementary Table 4 for a list of plasmids used). Two plasmids, pET29a(+)::*D1141A*

and pET29a(+)::*D1164A*, code for single point mutations at residues D1141 and D1164 in Tse5, which were mutated to alanine. Another plasmid, pET29a(+)::*K47G-P48A*, contains mutations for residues K47 and P48 to glycine and alanine, respectively. The pET29a(+)::Tse5-*ΔCT* plasmid encode a truncated version of Tse5, lacking Tse5-CT toxin.

*Escherichia coli* Lemo21(DE3) cells were transformed with each plasmid and grown at 37 °C in LB agar medium supplemented with 50 μg/mL kanamycin, 34 μg/mL chloramphenicol and 2 mM rhamnose. Protein overexpression was achieved by removing rhamnose from the LB medium. When cells reached OD$_{600}$ value of *ca.* 0.8-0.9, protein expression was induced by adding 1 mM isopropyl β-D-1-thiogalactopyranoside (IPTG), and cultures were left *ca.* 18 h at 18 °C. Then, cells were harvested and frozen for later use.

Cell pellet from 2 L culture was resuspended in 30 mL of 50 mM Tris-HCl pH 8.0, 500 mM NaCl, 20 mM imidazole and 2 μL of benzonase endonuclease and one tablet of protease inhibitor cocktail (cOmplete, EDTA-free, Roche). After disrupting cells by sonication, the suspension was centrifuged for 40 min at 43,000 xg. The soluble fraction was filtered with a 0.2 μm syringe filter and subjected to a first purification step at 4 °C by immobilised metal affinity chromatography using a HisTrap HP column of 5 mL (GE Healthcare) equilibrated with 25 mL of 50 mM Tris-HCl pH 8, 500 mM NaCl and 20 mM imidazole (solution A). The column was washed with solution A at 0.2 mL/min until no change in absorbance at 280 nm was detected. Elution was performed on a fast protein liquid chromatography system (ÄKTA FPLC; GE Healthcare) with a linear gradient between 0 and 50% of 50 mM Tris-HCl pH 8, 500 mM NaCl and 500 mM imidazole (solution B) in 40 mL and 2 mL/min. Fractions containing protein were pooled, and 2 mM DTT was added to the sample before injecting it into a HiLoad Superdex 200 26/600 pg, previously equilibrated in 20 mM Tris-HCl pH8, 150 mM NaCl and 2 mM DTT (Superdex buffer). Fractions containing the protein (as seen by SDS-PAGE, Supplementary Data 3) were pooled, concentrated using Amicon centrifugal filter units of 30 kDa molecular mass cut-off (Millipore) to a final concentration of *ca.* 2.5 mg mL$^{-1}$, flash frozen with liquid nitrogen and stored at −80 °C until use.

## Small-angle X-ray scattering study of Tse5, Tse5-ΔCT, Tse5-D1141A, and Tse5-K47G-P48A

Synchrotron X-ray scattering data for purified wild-type Tse5 and variants Tse5-K47G-P48A, Tse5-D1141A and Tse5-ΔCT were collected on an EigerX 4 M (Dectris) pixel detector at Diamond Light Source B21 beamline (UK). Data collection and structural parameters are reported in Supplementary Table 3. The scattering patterns were measured with a 3-s exposure time per frame in a continuous mode using an in-line Agilent HPLC system connected to a Shodex KW-403 column (exclusion limits: 10−700 kDa), equilibrated in 50 mM pH 8.0, 150 mM NaCl and 2 mM DTT, and running at 0.16 ml·min$^{-1}$. Proteins were injected at 2.5 mg mL$^{-1}$.

To check for radiation damage, the accumulated frames corresponding to eluted peaks were compared as a time series, and no radiation damage was observed. Using the sample-to-detector distance of 3.72 m, the range of momentum transfer values is $0.0045 < q < 0.34$ Å$^{-1}$ ($q = 4\pi\cdot\sin(\theta)/\lambda$ where $2\theta$ is the scattering angle, and $\lambda = 1$ Å is the X-ray wavelength).

Data were processed using standard procedures by the programme packages ScÅtter IV and PRIMUS 3.1[47]. The forward scattering [*I*(0)] was evaluated using the Guinier approximation[48] assuming the intensity is represented as $I(q) = I(0)\exp(-(qR_g)^{2/3})$ for a minimal range of momentum transfer values ($q < 1.3/R_g$). The maximum dimensions ($d_{max}$), the interatomic distance distribution functions [*P*(*r*)], and the radii of gyration ($R_g$) were computed using GNOM[49]. The molecular mass was estimated by a Bayesian approach[50] (a concentration-independent method), providing a probability estimate and credibility interval of the particles.

## Cryo-EM sample vitrification, data collection, image processing, structure determination, model building, refinement and validation of Tse5

For cryo-EM grid preparation, an extra purification step was performed. Freshly purified sample was concentrated up to *ca.* 20 mg/ml, and 0.5 ml was injected in a Superdex Increase 200 10/300 GL, previously washed and equilibrated in Superdex buffer. Fractions of 0.5 ml were collected, and the elution-peak was used to prepare grids after adding 0.05% CHAPS, which was essential to avoid the preferred orientation. To solve the prefered orientation problem, we also tried data collection on grids with an ultrathin carbon layer, and grid preparation using the Chameleon freezing device at the UK's national Electron Bio-imaging Centre (eBIC, Diamond Light Source).

UltrAuFoil R 1.2/1.3 300 mesh grids (Supplementary Data 3) were glow discharged at 0.36 mbar vacuum for 2 min. Then, 4 μL of purified Tse5 at 5 mg/ml with 0.05% CHAPS were applied to the grids and blotted for 2.3 s in a Leica EM GP2 single-side blotting automated plunge freezer at 94% humidity. Immediately after, grids were plunge-frozen in liquid ethane and stored in liquid nitrogen.

Preliminary data was collected in house on a 300 kV Thermo-Fisher Titan Krios G4 transmission electron microscope, paired with Gatan's BioContinuum Imaging Filter. 14,121 movies were recorded on a K3 direct electron detector device at a nominal magnification of 130,000 x with a calibrated pixel size of 0.6462 Å. A defocus range of −0.6 to −2 μm was used, with a total dose of 48 e$^-$/Å$^2$ fractionated over 50 frames with a total exposure time of 0.86 s. Acquired image stacks were processed using RELION 4.0 software[51], resulting in a final map that allowed to build the atomic structure of Tse5 at 3.28 Å.

The final data set was collected in a 300 kV Thermo-Fisher Titan Krios transmission electron at the UK's national Electron Bio-imaging Centre (eBIC, Diamond Light Source). 10,244 movies were recorded on a Falcon 4 direct electron detector at a nominal magnification of 130,000 x with a calibrated pixel size of 0.921 Å. A defocus range of −1 to −2 μm was use, with a total dose of 60 e$^-$/Å$^2$ in 2611 fractions during 10,84 s (dose rate of 5.5e$^-$/Å$^2$/s). Data collection details are described in Supplementary Fig. 1.

Acquired image stacks were processed using cryoSPARC v4[52], which yield a final 2.45 Å resolution map. Movie frames were aligned using cryoSPARC's Patch motion correction (multi) and contrast transfer function (CTF) for each aligned micrograph was estimated using Patch CTF estimation (multi). Based on CTF and total motion, the best micrographs were classified for particle selection. 2D classifications were performed on particles selected with the Block picker to generate templates for particle picking. After additional 2D classifications, initial models were obtained using ab initio reconstruction. The best model and 3 bad classes were subsequently *Hetero* refined to remove poorly aligned particles, and the remaining particles were *Homogeneously* refined using a tight mask. Further CTF and Defocus refinement were performed, followed by a 3D classification without alignment. 323,963 particles were selected for a *Homogeneous* Refinement before a Local Refinement, yielding a 2.45 Å map. A post-process map was obtained with DeepEMhancer for better visualisation and model-building purposes[53]. The 3DFSC online server was used to evaluate the final map quality, and the local resolution was estimated using cryoSPARC[54]. A schematic representation of the processing workflow is shown in Supplementary Fig. 1.

Automated ab initio model building was achieved for c.a. 80% of the structure with Buccaneer[55] building software from the CCP-EM package[56]. Guided by the DeepEMHancer and a standard post-processed map, several rounds of manual building and adjustment were performed with Coot 0.9.8.91[57] and subsequently refined with the Real-Space-Refine tool from the Phenix 1.20.1-4487 package[58]. Model validation was performed with MolProbity and the PBD validation service[59,60]. Final model statistics are summarised in Supplementary Table 1. Estimated global resolution and a local resolution map are

shown in Supplementary Fig. 2. Figures were prepared using ChimeraX[61].

### Study the partitioning in lipid monolayers of Tse5, Tse5-ΔCT, Tse5-D1141A, and Tse5-K47G-P48A

The capacity of wild-type Tse5, Tse5_K47G-P48A, Tse5_D1141A and Tse5_ΔCT to penetrate lipid monolayers was assessed by measuring its critical lateral pressure ($\Pi_c$) using the Langmuir–Blodgett balance technique with a DeltaPi-4 Kibron tensiometer (Helsinki, Finland). Each experiment was performed in a fixed-area circular trough (Kibron μTrough S system, Helsinki, Finland) of 2 cm in diameter, where 1.25 mL of the aqueous phase was added (5 mM Hepes pH 7.4, 150 mM NaCl). The temperature of the Langmuir balance was controlled thermostatically by a water bath at 25 °C (JULABO F12). The monolayer was formed by spreading over the aqueous surface *E. coli* polar lipid extract (Avanti Polar lipids, Supplementary Data 3) dissolved in chloroform at 1 mg/mL with a Hamilton microsyringe until the desired initial monolayer surface pressure was reached ($\Pi_0$). Experiments at different initial surface pressure ($\Pi_0$) values were recorded by changing the amount of lipid applied to the air-water interface ($\Pi_0$ value ranging from 15 to 30 mN/m). Then wild-type Tse5 or variants dissolved in 20 mM Tris·HCl pH 8.0, 150 mM NaCl, 2 mM DTT were injected into the aqueous subphase (final concentration of 0.4 μM), while controls were carried out by injecting buffer alone. Changes in surface pressure were monitored over time and were plotted as a function of $\Pi_0$. These data were fitted to a linear regression model, and the maximum insertion pressure was determined by extrapolation ($y$ value when $x = 0$).

### Electrophysical study of the pore-forming activity for Tse5, Tse5-CT, Tse5-D1141A, Tse5-K47G-P48A, and Tse5-ΔCT in planar lipid bilayers

Planar lipid membranes were formed by using a solvent-free modified Montal-Mueller technique[35]. Lipid was prepared from chloroform solutions of either a natural polar extract from *E. coli*, pure dioleoyl-phosphatidylethanolamine (DOPE), or a mixture of dioleoyl-phosphatidylethanolamine and dioleoyl-phosphatidylglycerol 67:33 w/w. All lipids were purchased from Avanti Polar Lipids (Supplementary Data 3). *E. coli* polar lipid extract headgroup composition is 67% phosphatidylethanolamine, 23.2% phosphatidylglycerol, and 9.8% cardiolipin, and acyl chains are the mixture naturally present in *E. coli*[62]. All lipids were dissolved in pentane at 5 mg/ml concentration after chloroform evaporation. Two monolayers were made by adding 10–30 μL of the lipid solution at each compartment of a Teflon chamber (so-called *cis* and *trans*), each filled with 1.8 ml salt solutions. The two compartments were partitioned by a 15 μm thick Teflon film with a *ca.* 100 μm diameter orifice where the bilayer formed. The orifice was pre-treated with a 3% solution of hexadecane in pentane. After pentane evaporation, the level of solutions in each compartment was raised above the orifice so the planar bilayer could form by apposition of the two monolayers. Capacitance measurements monitored correct bilayer formation. After bilayer formation, Tse5 or variants dissolved in 20 mM TRIS pH 8, 150 mM NaCl, and 2 mM DTT were added to the *cis* compartment to a final concentration of 168 nM.

To carry out the electrical measurements, an electric potential was applied using in house prepared Ag/AgCl electrodes in 2 M KCl, 1.5% agarose bridges assembled within standard 250 μl pipette tips. The potential is defined as positive when it is higher at the side of the protein addition (the *cis* side) while the *trans* side is set to ground. An Axopatch 200B amplifier (Molecular Devices, Sunnyvale, CA) in the voltage-clamp mode was used for measuring the current and applying potential. Data were filtered by an integrated low-pass 8-pole Bessel filter at 10 kHz, saved with a sampling frequency of 50 kHz with a Digidata 1440 A (Molecular Devices, Sunnyvale, CA), and analysed using pClamp 10 software (Molecular Devices, Sunnyvale,

CA). The membrane chamber and the head stage were isolated from external noise sources with a double metal screen (Amuneal Manufacturing Corp., Philadelphia, PA). The described setup can resolve currents of the order of picoamperes with a time resolution below the millisecond.

Current measurements were performed with a concentration gradient of 250 mM KCl *cis* / 50 mM KCl *trans* or 50 mM KCl *cis* / 250 mM KCl *trans*. All solutions were buffered with 5 mM HEPES at pH 7.4. The pH was adjusted by adding HCl or KOH and controlled during the experiments with a GLP22 pH meter (Crison). Steady current at each applied potential was calculated from a single Gaussian fitting of histograms of current values.

Selectivity measurements were performed during the experiments. Once the protein was inserted under the concentration gradient, a net ionic current appeared due to the existence of one or several selective pores. Selectivity was quantified by measuring the reversal potential (RP), corresponding to the applied voltage needed to cancel the current. If the channel is neutral, RP equals zero, while it becomes non-zero when the channel is selective to anions or cations. When the concentration gradient is 250/50 mM, a negative RP corresponds to cation-selective channels. RP was obtained from either the linear regression of the measured IV curves or by manually cancelling the observed current. All RP values were corrected by the liquid junction potential from Henderson's equation to eliminate the contribution of the electrode's agarose bridges[63]. Then, permeability ratios, $P_K^+/P_{Cl}^-$, were calculated from RP values using the GHK equation[64].

### DNA manipulation and mutant strain generation

All primers were synthesised and purified by Integrated DNA Technologies (Supplementary Data 3). Phusion polymerase, restriction enzymes, and T4 DNA ligase were purchased from New England Biolabs (NEB). Sanger sequencing was performed by the Centre for Applied Genomics (TCAG) at the Hospital for Sick Children (Toronto, Canada).

Chromosomal point mutants in *P. aeruginosa* were generated by double-crossover allelic exchange as previously described[65]. Briefly, approximately 500 bp flanks upstream and downstream of the desired mutation were amplified by PCR and spliced together by overlap extension PCR. Point mutations were engineered into the overlap between flanks. The resulting amplicon was ligated into the pEXG2 allelic exchange vector by digesting with HindIII and XbaI, transformed into SM10 and introduced into *P. aeruginosa* by conjugation. Merodiploids were selected on LB agar containing 30 μg/mL gentamicin and 25 μg/mL irgasan and streaked on LB agar lacking NaCl that contained 5% (w/v) sucrose for *sacB* counterselection. Strains that grow on sucrose and are gentamicin sensitive were screened by colony PCR using a primer that annealed specifically to the mutated nucleotides. Mutants were confirmed by PCR amplification of the appropriate region followed by Sanger sequencing of the resulting amplicon.

### Bacterial competition assays

Intraspecies competition assays were conducted as previously described with minor modifications[66]. Briefly, overnight cultures of indicated PAO1 strains were grown in LB at 37 °C in a shaking incubator. Cultures were normalised to an $OD_{600}$ of 1.0, and donor and recipient strains were mixed in a ratio of 5:1. Donor/recipient mixtures were spotted (10 μL) on a nitrocellulose membrane overlaid on LB 3% agar (w/v), and competitions were allowed to grow for 20 h at 25 °C. Competitions were resuspended in 1 mL LB broth, and colony forming units (CFU) were enumerated by serial dilution on LB 1.5% (w/v) agar containing 30 μg/mL gentamicin for selection of prey strains, which harbour a plasmid conferring gentamicin resistance (pPSV35-CV). Data are presented as the surviving prey (CFU/mL) at 24 h.

## Membrane binding studies for Tse5, Tse5-D1141A, and Tse5-ΔCT in supported lipid bilayers using QCM-D

Quartz crystal microbalance with dissipation monitoring data were recorded with a Q-SENSE E4 system (Q-Sense, Sweden) connected to a peristatic pump. The supported lipid bilayers were formed on silicon oxide sensor crystals of 50 nm (Supplementary Data 3) by using a modified version of the solvent-assisted lipid bilayer (SALB) method[67]. This approach is very quick and efficient because lipids are directly dissolved in an organic compound without the need to prepare small unilamellar vesicles. The process consists of the deposition of lipid solution on a solid support, followed by an exchange of the organic solvent with water. The modification to the original method consists of the use of water rather than buffer to avoid salt crystal formation upon contact with high concentrations of alcohol.

Prior to the experiments, the tubing system was thoroughly cleaned with a 2% Hellmanex solution, MiliQ water and ethanol. Silica surfaces were bath sonicated first with 2% Hellmanex for 10 min, and then with pure ethanol for 10 min, combining MiliQ water rinses after each step. Sensors were subsequently dried with $N_2$ and oxidised for 10 min using a UV-ozone chamber (BioForce Nanosciences, Inc., Ames, IA) to remove any molecular contamination and to increase their hydrophilicity.

The experiments were performed at a flow rate of 0.05 mL/min (Ismatec IPC 4-channel peristaltic pump, Cole-Parmer GmbH, Germany) and at a constant temperature of 25 °C. All the buffers were degassed extensively to avoid any bubble formation. Once the cell was assembled, the system was equilibrated with MiliQ water until reaching a stable baseline for six overtone frequencies (3rd, 5th, 7th, 9th, 11th, 13th). To form the supported lipid bilayers (SLBs), isopropanol was pumped, and when the baseline was stable, 2 ml at 0.5 mg/ml of a 9:1 POPG/POPE lipid mixture dissolved in isopropanol was injected (Supplementary Data 3). Then, the system was rinsed with water until the frequency stabilised around −23 Hz and dissipation, a clear indicator of an efficient SLB formation[68]. The system was equilibrated in TBS buffer before injecting the protein at 1 µM for ca. 45 min and subsequently rinsed with TBS buffer. For data analysis, the seventh harmonic was chosen in order to ensure data robustness. A typical replicate for SLB formation is shown in Fig. 5b.

## Bioinformatic analysis of Tse5 homologues

A bioinformatic analysis was carried out to find Tse5 homologues with Foldseek[42], which performs structural alignment of Tse5 with millions of predicted AlphaFold structures and the full Protein Databank. Using a sequence identity threshold of 18.2%, and after duplicated or truncated sequences were deleted, we identified 33 homologues (Supplementary Data 1). Next, we classify homologues based on putative C-terminal toxicities. We identified 8 homologues with predicted DNase/RNase activity, 4 with predicted ADP-ribosyltransferase activity, one with predicted peptidoglycan hydrolase, one with double-stranded DNA cytidine deaminase, and one with a putative colicin activity (see Supplementary Fig. 6 for phylogenetic analysis of C-terminal fragments found in Tse5 homologues).

We then aligned Tse5 with homologues containing C-terminal toxin fragments of putative/known enzymatic functions (Supplementary Fig. 7a). This alignment allows identification of conservation/divergence of residues lining the interior and the exterior of the Tse5-Shell (Supplementary Fig. 7b). To conduct this analysis, we first calculated a consensus sequence between aligned sequences, using a consensus threshold of 65% identity. Then, we search for residues in the Tse5-Shell that diverge from the consensus sequence.

## Molecular dynamics simulation of the Tse5 interfacial membrane binding

First, we built an atomic model of Tse5 combining the 2.45 Å cryo-EM structure with unresolved residues modelled by AlphaFold[41]. The predicted residue positions included Predicted Helical Region 1 of Tse5-Shell (residues 873–909) and a portion of the Tse5-CT fragment (residues 1196–1317). Then, CHARMM-GUI[69,70] was employed to generate an atomic interfacial membrane binding model. CHARMM-GUI applied the PPM Server[43] to position Tse5 in the membrane. The lipid headgroup composition of the model lipid bilayer corresponds to the E. coli Polar Lipid Extract (Avanti Polar Lipids) employed for our biophysical studies (67% DOPE, 23% DOPG, 10% 18:1 Cardiolipin (TLCL1)). The membrane-protein system was fully hydrated and neutralised with 150 mM $K^+/Cl^-$ ions. The atomic protein-membrane-solvent model was built on a rectangular cell of dimension 17x17x19 nm and comprises Tse5 residues 30–1317, 69144 DOPE molecules, 24104 DOPG molecules, 17352 TLCL1molecules, 613 $K^+$ ions, 343 $Cl^-$ ions, and 556902 water molecules. Finally, the CHARMM[71] interface generated topology and input files for MD simulation with GROMACS[72] (CHARMM36m[73,74] force field; TIP3P[75] water model).

To ensure the system's stability, a minimisation and equilibration protocol[76] was employed, including an initial set of two NVT ensemble simulations followed by four more NPT ensemble simulations. The first three steps lasted 0.125 ns, while the last three lasted 0.5 ns, granting a total equilibration of 1.875 ns. The energy minimisation run was performed before the equilibration and lasted 5 picoseconds. Trajectory data was saved in xtc format. Root Mean Square Fluctuation (RMSF) values were computed for the Cα in each residue of Tse5 over the whole range of the MD trajectory (1 µs), using gmx rmsf. We grouped lipid headgroups and acyl chains separately using gmx make_ndx to calculate the mass density profile of the system with gmx density.

The molecular dynamics simulations were conducted using GROMACS 2020.4[77] on the University of the Basque Country High-Performance Computing service. To ensure computational efficiency, the Message Passing Interface (MPI) method was used to harness parallel computing over 800 ranks in the cluster, and the simulation was instructed to allocate 160 ranks for Particle Mesh Ewald (PME) electrostatics, a method used for calculating long-range interactions. Dynamic load balancing was also allowed to ensure uniform computational load distribution. ChimeraX[61] was used to visualise the trajectory, prepare images, and record movies.

## Statistics and reproducibility

Statistical analyses were performed using GraphPad Prism 9.5 and are detailed in the figure legends.

## Reporting summary

Further information on research design is available in the Nature Portfolio Reporting Summary linked to this article.

# Data availability

The authors declare that source data supporting the findings of this study are available within the paper and its supplementary information files. Atomic coordinates of the Tse5 structure have been deposited in the Protein Data Bank (accession code id: 8CP6). The cryo-EM map is available from the Electron Microscopy Data Bank (accession code id: EMD-16778). Supplementary Data 1–3 are available as supplementary files and contain a list of Tse5 homologues identified by Foldseek, the Tse5 protein sequence derived for structural and biophysical studies, and a list of essential materials employed, respectively. Data plotted in Figures are available as supplementary material in a Source Data file. Supplementary Movie 1 shows the 1 µs MD trajectory of the Tse5-membrane simulation. Supplementary Notes 1–5 are available in the Supplementary Information file and contain the bioinformatic analysis of Tse5 homologues (Note 1), LC-ESI-MS report (Note 2), and the N-terminal sequencing reports for Tse5 (Notes 3–5). Uncropped and unedited gel images are included in Supplementary Fig. 11. Source data are provided with this paper.

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

## Acknowledgements

We acknowledge the Diamond Light Source for access and support of the cryo-EM facilities at the UK's national Electron Bio-imaging Centre (eBIC) [under proposals EM BI23872 and BI28248], funded by the Wellcome Trust, MRC and BBRSC. Also, we want to acknowledge Dr Miriam Weckener's support for grid preparation with the chamaleon® instrument (SPT Labtech). Some of this work was performed at the Basque Resource for Electron Microscopy located at Instituto Biofisika (UPV/EHU, CSIC), supported primarily by the Department of Education and the Innovation Fund of the Basque Government, with additional support from Fundación Biofísica Bizkaia and MCIN with funding from European Union NextGenerationEU (PRTR-C17.I1). In particular, Dr David Gil-Carton, Dr Jorge P. López-Alonso, and Dr Carlos Fernandez-Rodriguez are acknowledged for their cryo-EM support. We acknowledge the FGCZ for the mass spectrometry analyses and the technical support (Functional Genomics Center Zurich (FGCZ), University/ETH Zurich). The authors thank the technical and human support provided by the HPC SGIker (UPV/EHU/ ERDF, EU). D.A.-J. acknowledges support by the MICIN Contract PID2021-127816NB-I00, Fundación Biofísica Bizkaia, the Basque

Excellence Research Centre (BERC) programme, and IT1745-22 of the Basque Government. A.G.-M. acknowledges the financial support received from the Spanish Ministry of Universities and the Grants for the requalification of the Spanish university system for 2021–2023, financed by the European Union-Next Generation EU-Margarita Salas Modality. I.U.-B. acknowledges support by MICIN Contracts PID2019-104423GB-I00 and PID2022-143177NB-I00. J.C.W. acknowledges support from a project grant from the Canadian Institutes of Health Research (PJT-175011). M.C. acknowledge the financial support received from the IKUR Strategy under the collaboration agreement between Ikerbasque Foundation and Fundación Biofísica Bizkaia on behalf of the Department of Education of the Basque Government. A.A. acknowledges support from the Spanish Ministry of Science and Innovation (Projects 2019-108434GB and PID2022-142795NB-I00 funded by MCIN/AEI/10.13039/501100011033), and Universitat Jaume I (project UJI-B2022-42). A.A and J.R.-P. acknowledge support from Generalitat Valenciana (project CIGRIS/2021/021). M.Q.-M. acknowledges support from the Spanish Ministry of Science and Innovation (Project IJC2018-035283-I funded by MCIN/AEI/10.13039/501100011033) and Universitat Jaume I (project UJI-A2020-21). J.C.W. is the Canada Research Chair in Molecular Microbiology and holds an Investigators in the Pathogenesis of Infectious Disease Award from the Burroughs Wellcome Fund.

## Author contributions

A.G.M., I.T., J.A.-A., M.Q.-M., J.C., C.V., M.Z., and J.R.-P. designed and performed the experiments presented and contributed to writing the manuscript. I.T., M.Q.-M., M.C., J.C.W., A.A., and I.U.-B., contributed to project management, assisted with the experimental plan and contributed to writing the manuscript. D.A.-J. designed the overall experimental plan for the manuscript, wrote the manuscript, and contributed to project management.

## Competing interests

The authors declare no competing interest.
