## [Peer Review File · Nature Communications]

Structural and functional insights into the delivery of a bacterial Rhs pore-forming toxin to the membraneReviewer #1 (Remarks to the Author):

Gonzalez-Magana et al. describe the atomic structure of the *P. aeruginosa* type VI secretion system toxin, Tse5. The 3.23 Å structure reveals that Tse5 is made up of three polypeptides that form a shell-like structure that encapsulates the toxic C-terminal fragment of Tse5 and identify. Authors also provide evidence that the Tse5 shell itself interacts with the lipid bilayer, pointing towards a mechanism whereby the encapsulated toxin is first deposited in the membrane by shell-membrane interaction, and subsequently inserted to form a transmembrane pore. Based on computational analysis, authors present a model of how the shell surface electrostatics locate the shell peripherally in the membrane. Moreover, authors provide evidence that Tse5 auto-proteolysis of the toxic C-terminal fragment is essential to yield a mature toxin, but not of the N-terminal plug, which sets it apart from other Rhs proteins such as Rhs1 and RhsA.

The manuscript provides a novel atomic map of a pore-forming toxin Tse5, and authors recognize a new potential method for engagement of the toxin with the membrane – both are significant discoveries. However, this reviewer would like to raise additional question to help strengthening some of the conclusions of the manuscript. Firstly, the formation of shell-like structures and association with the membrane has already been well documented. The novelty of the work stems mainly from solving the structure of the Tse5 full-length protein, yet the manuscript fails to compare in detail this structure with previously reported structures of other Rhs proteins e.g. Rhs1 and RhsA. Furthermore, the proposed mode of association of the protein (that lacks designated transmembrane modules) with the membrane prior proteolysis and CT-toxic domain delivery is still speculative and requires additional experimental work.

Major concerns:

The structural insight of the Tse5 protein is a valuable result on its own and should be exhausted more. As only the first 27 residues of Tse5-CT were solved, authors should aim to provide a greater structural insight into the toxin itself, including the interaction of the toxin with the shell lumen and the N- and C-terminal plugs if possible. This could potentially help to better understand the mechanism of toxin release from the protein shell domain. To achieve this, I suggest that authors either perform focus refinement in the lumen of the shell, signal subtraction of the shell or, ideally, solve the structure of the catalytic inactive N- and C-terminal variant e.g. K47G-P48A, D1141A or K47G-P48A, D1164A.

One important mechanistic aspect of the toxin delivery is the contribution of membrane contacts to the cleavage of the Tse5-CT toxic domain. I suggest that the authors test the Tse5-CT cleavage in the presence or absence of artificial membranes.

Also unclear is whether the Tse5-CT toxin exerts its function in the outer or inner leaflet of the bacterial outer-membrane. Early evidence suggests that expression of Tse5-CT reduced *E. coli* viability only when targeted to the periplasmic compartment of the cell (Whitney et al Mol Micro 2014). Thus, I suggest that authors discuss this hypothesis in the manuscript and adapt their model accordingly.

Authors should provide a detailed comparison of the Tse5 structure with Rhs1 and RhsA, including comparison of the surface electrostatics, RMSD and volume/toxin size differences. This would better highlight the differences between different members of the Rhs family and emphasise the novelty of the findings. Comparison of the conserved C- and N-terminal plug structures should also be included, and any potential differences highlighted. It raises an important question whether electrostatics of the Tse5 shell evolved to substitute for the lack of a membrane-binding module.

Minor corrections:

Supplementary sections 1,2 and 3 should be included in the results section of the main text.

Supplementary Section 2: Missing Å3.

References 7 and 30 are duplicated.

Line 53: Please define which secretory pathway.

Line 201: Rh1 should read Rhs1.

Line L112 mentions DPXGX19DPXG motif whereas Line 54 mentions DPXGX18DPXG motif, could authors correct discrepancy.

Reviewer #2 (Remarks to the Author):

This manuscript describes the structure and function of the Tse5 protein of *Pseudomonas aeruginosa* PAO1. This toxic protein is a type VI secretion system effector that was first shown to kill prey bacteria by Whitney et al. back in 2014. Tse5 has rearrangement hot spot (RHS) peptide repeats, which have previously been shown to fold into a large beta-sheet cocoon structure that surrounds the C-terminal toxic domain. There are four major findings in this report: 1) a 3.28 Å resolution cryo-EM structure (without C-terminal toxin domain), 2) computational analysis to show that the cocoon has a hydrophobic patch that may be functionally important, 3) that C-terminal autoproteolysis is required for toxin activity, and 4) that N-terminal autoproteolysis is not important for toxin activity. Overall, the work appears to be technically sound and the writing is clear. However, I don't think that the results are particularly novel or impactful for the field. The beta-cocoon structure for RHS repeats was first reported in *Nature* back in 2013 for insecticidal Tc toxins. Since then, there have been (at least) two type VI secretion RHS structures published, each showing the same cocoon architecture as reported here. The interesting thing about Tse5 is that it carries a pore-forming toxin in the cocoon, but the pore-forming domain isn't resolved in the authors' model. The C-terminal processing results are also a recapitulation of prior work done in other systems: Tc toxins (2013), *Aeromonas dhakensis* (2020), *Enterobacter cloacae* (2020) and *Pseudomonas protegens* (2022). Probably the most interesting observation is that the exterior surface of the cocoon has a hydrophobic patch. Unfortunately, the authors do not try to perturb this feature to experimentally test its importance for toxicity. In its current state, the manuscript doesn't really tell us anything we didn't already know about RHS.

Comments.

1. Line 33. This statement in the abstract implies that this report was the first to show that RHS are chaperone-like proteins. As outlined above, it's clear from a decade of work that RHS repeats are delivery devices.
2. Related to point 1 above (and as mentioned by the authors), RHS proteins have polymorphic toxin domains. There is no comment or discussion as to how Tse5 differs from other RHS proteins that contain enzymatic toxins. Is there something about the interior of the cocoon that is adapted for the hydrophobic pore-forming toxin in Tse5?
3. Lines 82-83. I'm not sure that I agree with this statement. The actual experiment lacks negative and positive controls for peripheral membrane proteins. The data in Fig. 2c don't include a cytosolic protein negative control and also lack a known peripheral membrane protein. Without this context, I cannot interpret the data beyond what the authors tell me. Depending on salt and pH, soluble cytosolic proteins will stably associate with liposomes. This experiment needs such controls.
4. Lines 136-139. The authors should provide some context for the numerical values cited for MIP. This is not a common assay or metric, so the values have no meaning for general readers. What are the MIP values for integral membrane proteins? For soluble proteins? For MinD, which has a C-terminal helix that inserts into just one leaflet of the bilayer?

5. Lines 141-142. The assertion that the hydrophobic surface of the cocoon mediates membrane association would be stronger if the authors used mutagenesis to alter the surface charge and repeated the experiment. Additionally, the surface appears to be largely electronegative. At face value, it seems as though this would hinder the initial association through charge-charge repulsion with negatively charged phospholipids.

6. Line 201. Typographical error, should be "Rhs1".

7. Lines 255-262. There are many pore-forming colicins and eukaryotic toxins that are soluble, yet can insert into the membrane bilayer. The authors could have used this space to compare and contrast with RHS.

REVIEWER COMMENTS

Reviewer #1 (Remarks to the Author):

Gonzalez-Magana et al. describe the atomic structure of the *P. aeruginosa* type VI secretion system toxin, Tse5. The 3.23 Å structure reveals that Tse5 is made up of three polypeptides that form a shell-like structure that encapsulates the toxic C-terminal fragment of Tse5 and identify. Authors also provide evidence that the Tse5 shell itself interacts with the lipid bilayer, pointing towards a mechanism whereby the encapsulated toxin is first deposited in the membrane by shell-membrane interaction, and subsequently inserted to form a transmembrane pore. Based on computational analysis, authors present a model of how the shell surface electrostatics locate the shell peripherally in the membrane. Moreover, authors provide evidence that Tse5 auto-proteolysis of the toxic C-terminal fragment is essential to yield a mature toxin, but not of the N-terminal plug, which sets it apart from other Rhs proteins such as Rhs1 and RhsA.

The manuscript provides a novel atomic map of a pore-forming toxin Tse5, and authors recognize a new potential method for engagement of the toxin with the membrane – both are significant discoveries. However, this reviewer would like to raise additional question to help strengthening some of the conclusions of the manuscript. Firstly, the formation of shell-like structures and association with the membrane has already been well documented. The novelty of the work stems mainly from solving the structure of the Tse5 full-length protein, yet the manuscript fails to compare in detail this structure with previously reported structures of other Rhs proteins e.g. Rhs1 and RhsA. Furthermore, the proposed mode of association of the protein (that lacks designated transmembrane modules) with the membrane prior proteolysis and CT-toxic domain delivery is still speculative and requires additional experimental work.

Major concerns:

The structural insight of the Tse5 protein is a valuable result on its own and should be exhausted more. As only the first 27 residues of Tse5-CT were solved, authors should aim to provide a greater structural insight into the toxin itself, including the interaction of the toxin with the shell lumen and the N- and C-terminal plugs if possible. This could potentially help to better understand the mechanism of toxin release from the protein shell domain. To achieve this, I suggest that authors either perform focus refinement in the lumen of the shell, signal subtraction of the shell or, ideally, solve the structure of the catalytic inactive N- and C-terminal variant e.g. K47G-P48A, D1141A or K47G-P48A, D1164A.

R1: Thanks for the comment. Following the reviewer's suggestion, we have performed focus refinement to improve the cryo-EM density inside the shell using Relion 4.0. To this end, we generated masks for the densities in the inner part of the shell, which include the residues of the N- and C-terminal plugs (residues 49-124 and 1092-1168), the residues of the N- and C-terminal fragments (Tse5-NT residues 30-48, and Tse5-CT residues 1169-1195), and two regions of 10 and 22 residues respectively for which we built polyalanine fragments that are located inside the Tse5-Shell and that most likely belong to Tse5-CT residues not assigned. Despite we were able to cover the lumen of the Tse5-Shell by softening and extending the

masks, the results of the focus refinements did not extend the inner density, and we were unable to assign more Tse5-CT residues.

Since we could not improve the density inside the shell using the original data, we have collected more cryo-EM data increasing sample concentration to yield more particles per micrograph. The new cryo-EM data collected at eBIC (Diamond Light Source, UK) improved the overall resolution from 3.2 to 2.45 Å.

The new map showed more featured densities inside the barrel. However, we could not build more residues of Tse5-CT due to a lack of connectivity between them. The densities that have not been assigned can be seen in Supplementary Fig. 3a.

To improve the connectivity between unassigned densities, we tried focus refinement on the new data using cryoSPARC and masks covering the lumen of the shell. Unfortunately, we did not manage to improve the connectivity between densities, and therefore no more residues of Tse5-CT were built.

Even after improving the overall resolution from 3.2 to 2.5 Å, we can only unambiguously assign 27 residues out of 147 in Tse5-CT, suggesting that most of this fragment is flexible/disordered inside the Tse5 shell. Flexibility/disorder might also explain why only 10% and 23% of the C-terminal toxin fragments in the cryo-EM structures of Rhs1 [1] and RhsA [2] were resolved, respectively.

The revised manuscript has been updated with figures showing the new cryo-EM map and rebuild/refined structure. Also, the Methods have been updated with new cryo-EM experimental details. New cryo-EM data collection, processing and model refinement statistics are presented in Supplementary Table 1.

One important mechanistic aspect of the toxin delivery is the contribution of membrane contacts to the cleavage of the Tse5-CT toxic domain. I suggest that the authors test the Tse5-CT cleavage in the presence or absence of artificial membranes.

R2: Thanks for the comment and suggestion. The data available in the literature points towards a conserved auto-proteolytic mechanism that results in the cleavage of the Rhs C-terminal fragments. This cleavage is mediated by the conserved aspartyl protease active site comprised of a DPXGX₁₈DPXG motif. Single-point mutants of conserved aspartic residues in Rhs proteins abolish the autoproteolytic activity, which is also true for Tse5 point mutants Tse5-D1141A and Tse5-D1164A.

Importantly, the available data indicate that auto-proteolytic cleavage occurs before secretion and is independent of T6SS functionality [3,4].

In particular, previous studies by the laboratory of Tao Dong [3] on the *Acidovorax citrulli* T6SS effector RhsB have tested its auto-proteolytic activity using the PURExpress® In Vitro Protein Synthesis kit consisting of defined transcription-translation protein components. Western blot analysis of *in vitro* expressed RhsB confirms that RhsB is self-cleaved at the C-terminus. In addition, they detected RhsB cleavage in the cytosol of *A. citrulli* by expressing its nontoxic mutant RhsB^{KE-AA} in *A. citrulli* wild type, T6SS-null mutant Δ tssM, and Δ rhsB, suggesting that self-cleavage of RhsB could occur before secretion and be independent of T6SS functionality [3].

The same laboratory has also studied the auto-proteolytic cleavage of the *Aeromonas dhakensis* T6SS Rhs effector Tsel [4] using the same commercial in vitro protein synthesis kit to express Tsel variants flanked by N-FLAG and C-3V5 epitope tags. Western blot analysis shows that Tsel was cleaved similarly in vitro, and mutations of D1407A and E428A also abolished cleavage, suggesting that Tsel is self-cleaved [4].

Given that C-terminal cleavage of RhsB and Tsel occurs in the absence of membranes using the PURExpress® In Vitro Protein Synthesis Kit (New England BioLabs), and that the conserved aspartyl protease active site mediates it, it seems reasonable to expect that membrane contacts are not necessary for Tse5-CT cleavage, therefore testing Tse5-CT cleavage in the presence or absence of artificial membranes would not seem necessary. Nevertheless, if the reviewer thinks we should perform the *in vitro* protein synthesis assay with and without membranes, we could afford it.

In the revised manuscript, we have included the available literature regarding the *in vitro* auto-proteolytic function of RhsB and Tsel proteins (lines 251-256).

Also unclear is whether the Tse5-CT toxin exerts its function in the outer or inner leaflet of the bacterial outer-membrane. Early evidence suggests that expression of Tse5-CT reduced *E. coli* viability only when targeted to the periplasmic compartment of the cell (Whitney et al Mol Micro 2014). Thus, I suggest that authors discuss this hypothesis in the manuscript and adapt their model accordingly.

R3: Thanks for the suggestion, it is very much appreciated. Two laboratories [5,6], almost simultaneously, discovered Tse5. As indicated, Whitney et al. demonstrated that Tse5-CT is toxic when directed to its periplasm [6], and Hachani et al. show it was toxic when expressed in the cytoplasm of *Escherichia coli* [5]. Furthermore, we also corroborated the above results [7], showing that ectopic expression of Tse5-CT or a variant encoding for the PelB leader sequence (sp-Tse5-CT) in *Pseudomonas putida* EM383 cells leads to cell depolarisation.

Furthermore, we have shown that Tse5-CT forms ion-selective membrane pores when reconstituted on a planar lipid bilayer [7]. In this experimental setup, we assemble a planar lipid bilayer that separates two chambers (*cis* and *trans*-sides). Upon addition of Tse5-CT in the *cis*-side, we observed spontaneous Tse5-CT insertion into the lipid bilayer, characterized by ion channel activity with relatively stable currents [7]. Tse5-CT-induced currents were obtained using membranes formed with a polar lipid extract from *E. coli* in a 250/50 mM or a 50/250 mM KCl gradient. Protein was always added at the same side of the membrane (*cis*-side), meaning that the gradient direction did not affect the capacity of Tse5-CT to insert into the planar membrane. This result is in agreement with the capacity of Tse5-CT to be active when directed to its periplasm and also when expressed in the cytoplasm. Supporting the idea that Tse5-CT can spontaneously insert the inner membrane of Gram-negative bacteria, either from its outer or inner leaflet.

But the question remains if Tse5 can also deliver Tse5-CT when approaching the inner membrane from its cytosolic or periplasmic site. To address this question, in the revised version of the manuscript, we present a detailed biophysical study of Tse5 pore-forming activity as a function of the salt concentration gradient and asymmetry in the lipid composition of the membrane. The aim of these experimental setups is to emulate the

electrical potential across the membrane. Interestingly, our data suggest that Tse5 might react differently depending on the orientation of the electrical potential across the membrane. In particular, our findings suggest that a negative transmembrane potential could inhibit Tse5's ability to deliver its toxic cargo.

Given that this potential corresponds to the one felt from the cytoplasmic side of the cell membrane, we postulate that Tse5 preferably acts from the periplasmic side. All this new data is introduced and discussed in detail in a new section (Tse5 delivers its encapsulated Tse5-CT toxin to target membranes). Also, following the reviewer's suggestion, in the revised manuscript, we have discussed all the previous results in the Introduction (lines 80-87) and updated our model accordingly (Fig. 6).

Authors should provide a detailed comparison of the Tse5 structure with Rhs1 and RhsA, including comparison of the surface electrostatics, RMSD and volume/toxin size differences. This would better highlight the differences between different members of the Rhs family and emphasise the novelty of the findings. Comparison of the conserved C- and N-terminal plug structures should also be included, and any potential differences highlighted. It raises an important question whether electrostatics of the Tse5 shell evolved to substitute for the lack of a membrane-binding module.

R4. Thanks for the suggestion. To address the reviewer's comment, we made several changes to the manuscript:

1. We include a new figure in Supplementary Information (Supplementary Fig. 8) showing the electrostatic surface potential of Tse5, Rhs1 and RhsA.
2. RMSD's, volumes and sizes of Tse5, Rhs1 and RhsA have been included in sections "Tse5 structural insight from its 2.45 Å cryo-EM structure" and "Differential requirement of Tse5-CT and Tse5-NT cleavage for toxin activity."
3. We include a new figure (Fig. 2a) showing a comparison between the conserved C- and N-terminal plug structures. The new figure is discussed in the section: "Differential requirement of Tse5-CT and Tse5-NT cleavage for toxin activity."
4. We include a new figure (Fig. 5a) showing an overall comparison between Tse5, Rhs1 and RhsA. The main differences are discussed in the section "A model of interfacial membrane binding for Tse5-CT toxin delivery."
5. We include a bioinformatic analysis to study the evolution of the Tse5-Shell surface with respect to other Rhs toxins containing membrane-binding modules in their N-terminal fragment (see Supplementary Information (Bioinformatic analysis of Tse5 homologues)). The most relevant information from this bioinformatic analysis has been included in section "A model of interfacial membrane binding for Tse5-CT toxin delivery."
6. Also, we introduced another biophysical technique to study the membrane-binding properties of Tse5 on supported lipid bilayers using Quartz Crystal Microbalance with Dissipation (QCM-D) monitoring, which is a highly sensitive analytical tool used to monitor changes in the mass and viscoelastic properties of thin films or surfaces in

real-time at the nanoscale. Using this technique, we evaluated the adsorption and desorption properties of Tse5 and mutants. The new data shows that Tse5 binding is largely irreversible, while, upon rinsing with buffer, the binding of Tse5- Δ CT or Tse5-D1141A yields a conformation of the adsorbed protein-lipid bilayer that significantly differs from that of the Tse5-lipid bilayer. This would indicate that the mutants can change conformation, probably due to missing interactions with the bilayer core. This new data provides evidence for the delivery mechanism of Tse5. These new experimental results are discussed in the section “Interfacial Tse5-membrane binding delivers Tse5-CT toxin to the target membrane.” Fig. 4 shows the QCM-D data.

Minor corrections:

Supplementary sections 1,2 and 3 should be included in the results section of the main text. Supplementary Section 2: Missing Å3.

R5. Thanks for the suggestion.

The previous Supplementary Sections 1-3 (Sequence variability of Tse5 YD-repeats, The cryo-EM map of Tse5-CT indicates it is partially flexible or disordered, and Protein-protein molecular interactions anchor Tse5-NT fragment to the Tse5-Shell fragment) have been included in the main text, section “Tse5 structural insight from its 2.45 Å cryo-EM structure main text.”

The missing “Å3” has been corrected.

References 7 and 30 are duplicated.

R6. The duplicate reference has been deleted.

Line 53: Please define which secretory pathway.

R7. We have defined several of the secretory pathways associated with Rhs trafficking, and included a reference for further information (lines 54-57).

Line 201: Rh1 should read Rhs1.

R8. The typo has been corrected.

Line L112 mentions DPXGX19DPXG motif whereas Line 54 mentions DPXGX18DPXG motif, could authors correct discrepancy.

R9. In Line 112, DPXGX19DPXG refers to the aspartyl protease motif found in Tse5, whereas in Line 54, DPXGX18DPXG refers to the aspartyl protease motif found in Rhs proteins from the Enterobacteriaceae family [9]. Thus, in Tse5, this motif contains an extra residue between the catalytic aspartic residues. In the revised manuscript, we indicated this sequence divergence between Tse5 and other Rhs proteins from the Enterobacteriaceae family (Lines 239-241).

Reviewer #2 (Remarks to the Author):

This manuscript describes the structure and function of the Tse5 protein of *Pseudomonas aeruginosa* PAO1. This toxic protein is a type VI secretion system effector that was first shown to kill prey bacteria by Whitney et al. back in 2014.

Tse5 has rearrangement hot spot (RHS) peptide repeats, which have previously been shown to fold into a large beta-sheet cocoon structure that surrounds the C-terminal toxic domain. There are four major findings in this report: 1) a 3.28 Å resolution cryo-EM structure (without C-terminal toxin domain), 2) computational analysis to show that the cocoon has a hydrophobic patch that may be functionally important, 3) that C-terminal autoproteolysis is required for toxin activity, and 4) that N-terminal autoproteolysis is not important for toxin activity. Overall, the work appears to be technically sound and the writing is clear. However, I don't think that the results are particularly novel or impactful for the field. The beta-cocoon structure for RHS repeats was first reported in *Nature* back in 2013 for insecticidal Tc toxins. Since then, there have been (at least) two type VI secretion RHS structures published, each showing the same cocoon architecture as reported here. The interesting thing about Tse5 is that it carries a pore-forming toxin in the cocoon, but the pore-forming domain isn't resolved in the authors' model. The C-terminal processing results are also a recapitulation of prior work done in other systems: Tc toxins (2013), *Aeromonas dhakensis* (2020), *Enterobacter cloacae* (2020) and *Pseudomonas protegens* (2022). Probably the most interesting observation is that the exterior surface of the cocoon has a hydrophobic patch. Unfortunately, the authors do not try to perturb this feature to experimentally test its importance for toxicity. In its current state, the manuscript doesn't really tell us anything we didn't already know about RHS.

Comments.

Line 33. This statement in the abstract implies that this report was the first to show that RHS are chaperone-like proteins. As outlined above, it's clear from a decade of work that RHS repeats are delivery devices.

R10. Thanks for pointing this out to us; it is very much appreciated. Following the reviewer's comment, we have rewritten the **ABSTRACT** to clarify that our findings relate to Tse5's function.

Related to point 1 above (and as mentioned by the authors), RHS proteins have polymorphic toxin domains. There is no comment or discussion as to how Tse5 differs from other RHS proteins that contain enzymatic toxins. Is there something about the interior of the cocoon that is adapted for the hydrophobic pore-forming toxin in Tse5?

R11. Thanks for the suggestion. We have performed a bioinformatic analysis to study the evolution of the Tse5-Shell/cocoon's interior and exterior surfaces with respect to other Rhs toxins containing known/predicted enzymatic C-terminal toxins (see Supplementary Information (Bioinformatic analysis of Tse5 homologues)). The most relevant information from this bioinformatic analysis has been included in the main text, section: "A model of interfacial membrane binding for Tse5-CT toxin delivery."

Regarding the adaptation of the interior of the cocoon for encapsulating the hydrophobic pore-forming toxin in Tse5, our bioinformatic analysis points to regions on the inner surface

of the cocoon that accumulate mutations and, therefore, could be relevant for its specialisation. We identify 32 residues lining the interior of the Tse5 cocoon that diverge from the consensus sequence (Suppl. Table 6). Seven mutations introduce a negative-charged residue, three introduce a positive-charged residue, and 14 introduce small nonpolar residues. Furthermore, 8 mutations substitute a hydrophobic residue with a small nonpolar or charged residue. Overall, these mutations modulate the electrostatic inner-surface potential, which could be important for the specialisation of Tse5 towards encapsulating and delivering a hydrophobic pore-forming toxin. Nonetheless, given that only 27 residues of the Tse5-CT fragment were resolved in the cryo-EM map, we cannot identify if divergent residues in the interior surface of the cocoon mediate protein-protein interactions with Tse5-CT.

We would like to emphasise that we made a considerable effort to improve the cryo-EM density inside the shell (Please, see **R1** above for details). To this end, we have performed focused refinement and also collected new cryo-EM data, which improved the overall resolution from 3.2 to 2.45 Å. Nevertheless, we can only unambiguously assign 27 residues out of 147 in Tse5-CT, suggesting that most of this fragment is flexible/disordered inside the Tse5 shell. Flexibility/disorder might also explain why only 10% and 23% of the C-terminal toxin fragments in the cryo-EM structures of Rhs1 [1] and RhsA [2] were resolved, respectively.

Lines 82-83. I'm not sure that I agree with this statement. The actual experiment lacks negative and positive controls for peripheral membrane proteins. The data in Fig. 2c don't include a cytosolic protein negative control and also lack a known peripheral membrane protein. Without this context, I cannot interpret the data beyond what the authors tell me. Depending on salt and pH, soluble cytosolic proteins will stably associate with liposomes. This experiment needs such controls.

R12. Thanks for the comment. It is very much appreciated. If we understand correctly, the reviewer questions our statement that the Tse5 cocoon is a peripheral membrane protein. So, we have rewritten this sentence to tone down the statement, and also, we avoid defining Tse5-Shell as a peripheral membrane protein (lines 94-99).

For a negative control with BSA in our Langmuir-Blodgett balance assay, please see response 15 (**R15**).

Regarding positive controls, the membrane insertion pressures (critical lateral pressure - Π_c) of many membrane-associated proteins/peptides have been reviewed in ref. [10]. This review highlights the diverse range of Π_c values for peripheral membrane proteins. For example, the Π_c values for glycolipid transfer protein (GLTP) can vary between 23 to 31 depending on the lipid charge. Thus, many factors modify Π_c values, including the experimental conditions (salt, pH), the membrane-binding mechanism, the presence of amphipathic helices, or lipid composition (Please, see **R13-15** for further details).

Importantly, the lipid packing in the outer monolayer of biological membranes approaches lateral surface pressures between 30 to 35 mN/m [10,11]. Thus, a critical lateral pressure in this range upon protein addition indicates that it can insert into the hydrophobic core of the lipid monolayer. If the Π_c values of a peripheral membrane protein are below 30 mN/m, it

indicates that its binding to the membrane is more superficial, and it is not inserted into the hydrophobic core of the lipid monolayer.

Given the many variables involved in Π_c measurements, reporting a single maximum insertion pressure (Π_c) value is not informative. Therefore, positive controls with other proteins are typically not included in these experiments. What is relevant is to observe changes in Π_c values depending on experimental conditions or protein variants. So, for example, in our case, we observe a significant reduction of the Π_c values obtained for Tse5-CT/Tse5 (~ 35 mN/m), while the Π_c value for Tse5- Δ CT drops to 27 mN/m (Fig. 3a). This dramatic reduction indicates Tse5- Δ CT is not able to insert into the hydrophobic core of a lipid monolayer.

Furthermore, in the revised manuscript, we provide further experimental data that supports the notion of interfacial Tse5-membrane binding for delivering Tse5-CT toxin to the membrane:

In particular, we introduced another biophysical technique to study the membrane-binding properties of Tse5 on supported lipid bilayers using Quartz Crystal Microbalance with Dissipation (QCM-D) monitoring. Using this technique, we evaluated the adsorption and desorption properties of Tse5 and mutants.

The new data shows that Tse5 binding is largely irreversible, while, upon rinsing with buffer, the binding of Tse5- Δ CT or Tse5-D1141A yields a conformation of the adsorbed protein-lipid bilayer that significantly differs from that of the Tse5-lipid bilayer. This would indicate that the mutants can change conformation, probably due to missing interactions with the bilayer core. This new data provides evidence for the delivery mechanism of Tse5. These new experimental results are discussed in the section “Interfacial Tse5-membrane binding delivers Tse5-CT toxin to the target membrane.” Fig. 4 shows the QCM-D data.

Lines 136-139. The authors should provide some context for the numerical values cited for MIP. This is not a common assay or metric, so the values have no meaning for general readers.

R13. Thanks for the comment. We have provided a better introduction to the Langmuir-Blodgett balance technique when we introduce it in the section “Tse5 delivers its encapsulated Tse5-CT toxin to target membranes.”

Also, the numerical context has been provided in the section “Interfacial Tse5-membrane binding delivers Tse5-CT toxin to the target membrane” when comparing the Π_c values we obtain for Tse5 and Tse5- Δ CT (lines 359-363).

What are the MIP values for integral membrane proteins?

R14. If correct insertion conditions are provided, the Π_c value of an integral membrane protein should be within the 30-35 mN/m range, given that it will insert into the hydrophobic core of the lipid monolayer in the Langmuir Balance. That is the case for the pore-forming Tse5-CT toxin, which contains at least a transmembrane region and has an Π_c value near 35 mN/m [7].

For soluble proteins?

R15. The concept of Π_c is primarily associated with inserting hydrophobic or amphipathic molecules into lipid bilayers. It is not typically used to describe the behaviour of soluble proteins. Nevertheless, we have obtained data for BSA to show what happens when a soluble protein is used in this assay. As we can see in the graph below, when a monolayer is assembled with an initial lateral pressure (Π_0) above ~ 23 mN/m, the $\Delta\Pi$ values start to be negative, which indicates the protein is unable to increase the lateral pressure of the monolayer, even though the monolayer has not reached a high-density packing of the lipids.

Langmuir Balance with BSA

For MinD, which has a C-terminal helix that inserts into just one leaflet of the bilayer?

R16. We have not found a bibliographic reference regarding the Π_c value for MinD. As pointed out by the reviewer, MinD's membrane-binding mechanism involves the insertion of its C-terminal amphipathic helix into one leaflet of the bilayer, and combines hydrophobic and electrostatic interactions [12].

The Π_c value for MinD will depend on how deep into the lipid monolayer it can reach. Thus, if MinD can insert into the hydrophobic core of the lipid monolayer, one would expect Π_c values to exceed the 30 mN/m threshold. On the other hand, if MinD-membrane interaction is more superficial (e.g., MinD binds at the interface between the lipid's polar heads and hydrophobic tails), one would expect an Π_c value below the 30 mN/m threshold. The latter scenario is what we observe for Tse5- Δ CT ($\Pi_c \sim 27$ mN/m).

82. Lines 141-142. The assertion that the hydrophobic surface of the cocoon mediates membrane association would be stronger if the authors used mutagenesis to alter the surface charge and repeated the experiment. Additionally, the surface appears to be largely electronegative. At face value, it seems as though this would hinder the initial association through charge-charge repulsion with negatively charged phospholipids.

R17. Thanks for the suggestion. Following the reviewer's comment, we have attempted mutagenesis to change the surface properties of Tse5. As shown in the figures below, the amphipathic surface of Tse5 is very large, containing many positively charged and hydrophobic residues. We have recalculated the electrostatic surface potential using the Coulombic equation (done in ChimeraX). Previous electrostatic surface potential was calculated by solving the Poisson-Boltzmann equation (using the APBS server[13]).

Given the large surface area that could be involved in protein-membrane interactions, and therefore affect the pore-forming activity of Tse5, we decided to mutate many of these residues to glutamine (mutated residues are shown in the figure below). The rationale behind this strategy was the assumption that only a substantial change in the physicochemical properties of this surface would result in an inhibition of its pore-forming activity. Unfortunately, most of the protein we obtained for this mutant is insoluble, and only a small fraction could be rescued (~0.1 mg of soluble protein per litre of culture). Furthermore, our quality controls indicate that it does not preserve its folding. Therefore, we are not confident in reporting these results.

Given that we have not experimentally demonstrated that the hydrophobic/amphipathic surface of the cocoon mediates membrane association, we have significantly reduced the strength of our claim. In particular, we reduce our claim to a model of interfacial membrane binding, and we do not discard the possibility that other surface regions in Tse5 might be involved in the interfacial Tse5-membrane binding for Tse5-CT toxin delivery.

We are basing our model of interfacial membrane binding on the structural comparison between Tse5, RhsA and Rhs1, the bioinformatic analysis of Tse5 homologues containing enzymatic C-terminal toxins, and a 1 μ s-MD simulation. Please, see the final section of the paper for details (A model of interfacial membrane binding for Tse5-CT toxin delivery).

Fig. 1 | Two views of the Tse5-Shell's amphipathic surface showing the Coulombic electrostatic potential plotted on the solvent-excluded surface (left panel), and residues on the surface contributing to its amphipathic character, including residues in *Pred. Helical Region 1*, *Helical Region 2*, *Hydrophobic Patch*, and other arginine residues on the surface. ChimeraX was used to calculate the Coulombic electrostatic potential.

83. Line 201. Typographical error, should be "Rhs1".

R18. Corrected. Line 301.

84. Lines 255-262. There are many pore-forming colicins and eukaryotic toxins that are soluble, yet can insert into the membrane bilayer. The authors could have used this space to compare and contrast with RHS.

R19. Thanks for the suggestion. We have compared the activities of pore-forming colicins in the final **SUMMARY**.

REFERENCES

1. Jurėnas D, Rosa LT, Rey M, Chamot-Rooke J, Fronzes R, Cascales E: **Mounting, structure and autocleavage of a type VI secretion-associated Rhs polymorphic toxin.** *Nat Commun* 2021, **12**:6998.
2. Günther P, Quentin D, Ahmad S, Sachar K, Gatsogiannis C, Whitney JC, Raunser S: **Structure of a bacterial Rhs effector exported by the type VI secretion system.** *PLoS Pathog* 2022, **18**:e1010182.

3. Pei T, Kan Y, Wang Z, Tang M, Li H, Yan S, Cui Y, Zheng H, Luo H, Liang X, et al.: **Delivery of an Rhs-family nuclease effector reveals direct penetration of the gram-positive cell envelope by a type VI secretion system in *Acidovorax citrulli*.** *mLife* 2022, **1**:66–78.
4. Pei T-T, Li H, Liang X, Wang Z-H, Liu G, Wu L-L, Kim H, Xie Z, Yu M, Lin S, et al.: **Intramolecular chaperone-mediated secretion of an Rhs effector toxin by a type VI secretion system.** *Nat Commun* 2020, **11**:1865.
5. Hachani A, Allsopp LP, Oduko Y, Filloux A: **The VgrG proteins are “à la carte” delivery systems for bacterial type VI effectors.** *Journal of Biological Chemistry* 2014, **289**:17872–17884.
6. Whitney JC, Beck CM, Goo YA, Russell AB, Harding BN, De Leon JA, Cunningham DA, Tran BQ, Low DA, Goodlett DR, et al.: **Genetically distinct pathways guide effector export through the type VI secretion system.** *Mol Microbiol* 2014, **92**:529–542.
7. González-Magaña A, Altuna J, Queralt-Martín M, Largo E, Velázquez C, Montánchez I, Bernal P, Alcaraz A, Albesa-Jové D: **The *P. aeruginosa* effector Tse5 forms membrane pores disrupting the membrane potential of intoxicated bacteria.** *Commun Biol* 2022, **5**:1189.
8. Graber ZT, Shi Z, Baumgart T: **Cations induce shape remodeling of negatively charged phospholipid membranes.** *Physical Chemistry Chemical Physics* 2017, **19**:15285–15295.
9. Jackson AP, Thomas GH, Parkhill J, Thomson NR: **Evolutionary diversification of an ancient gene family (rhs) through C-terminal displacement.** *BMC Genomics* 2009, **10**:584.
10. Calvez P, Bussièrès S, Éric Demers, Salesses C: **Parameters modulating the maximum insertion pressure of proteins and peptides in lipid monolayers.** *Biochimie* 2009, **91**:718–733.
11. Demel RA, Geurts van Kessel WSM, Zwaal RFA, Roelofsen B, van Deenen LLM: **Relation between various phospholipase actions on human red cell membranes and the interfacial phospholipid pressure in monolayers.** *BBA - Biomembranes* 1975, **406**:97–107.
12. Zhou H, Lutkenhaus J: **Membrane Binding by MinD Involves Insertion of Hydrophobic Residues within the C-Terminal Amphipathic Helix into the Bilayer.** *J Bacteriol* 2003, **185**:4326–4335.
13. Jurrus E, Engel D, Star K, Monson K, Brandi J, Felberg LE, Brookes DH, Wilson L, Chen J, Liles K, et al.: **Improvements to the APBS biomolecular solvation software suite.** *Protein Science* 2018, **27**:112–128.
14. Zhang D, de Souza RF, Anantharaman V, Iyer LM, Aravind L: **Polymorphic toxin systems: Comprehensive characterization of trafficking modes, processing, mechanisms of action, immunity and ecology using comparative genomics.** *Biol Direct* 2012, **7**:18.

Reviewer #1 (Remarks to the Author):

The authors have addressed my concerns satisfactorily. The revised manuscript includes enhanced maps and a more detailed section titled "Tse5 delivers its encapsulated Tse5-CT toxin to target membranes," supporting their proposed model of Tse5-CT interaction with the outer (periplasmic) membrane leaflet.

Tiago Costa

Reviewer #2 (Remarks to the Author):

My original criticisms focused on the lack of new insights into RHS structure and function, because two RHS structures have been published and N- and C-terminal autoprocessing reactions have already been described in the literature. In response to my question about differences between RHS proteins with cytoplasmic toxins versus the pore forming toxin of Tse5, the authors have now identified two alpha-helical elements on the outside of the Tse5 shell. These elements were not described in the original manuscript, and in the revision the authors describe molecular dynamics simulations suggesting that the helices penetrate into the membrane. The authors provide evidence that extra-shell helices are not found in RHS proteins that carry cytoplasmic toxins. These observations are the most interesting findings in the paper and the only data that justifies the manuscript title. However, it's puzzling that the authors did not go on to test the importance of the helices experimentally. They should delete each element individually and in combination and test the function of mutant Tse5 proteins in competitions. They should also test whether the helical deletions affect C-terminal autoprocessing. If the authors show that the helices are required for toxin insertion into the membrane, then the work would be a significant advance and worthy of publication in Nature Communications.